# Rosalia: An experimental research site to study hydrological processes in a forest catchment

Josef Fürst[1], Hans Peter Nachtnebel[1], Josef Gasch[2], Reinhard Nolz[3], Michael Paul Stockinger[3], Christine Stumpp[3], Karsten Schulz[1]

[1]Institute for Hydrology and Water Management (HyWa), University of Natural Resources and Life Sciences Vienna (BOKU), Vienna, Austria.
[2]Forest Demonstration Centre, University of Natural Resources and Life Sciences Vienna (BOKU), Vienna, Austria.
[3]Institute for Soil Physics and Rural Water Management (SoPhy), University of Natural Resources and Life Sciences Vienna (BOKU), Vienna Austria.

*Correspondence to*: Josef Fürst (josef.fuerst@boku.ac.at)

## Abstract

Experimental watersheds have a long tradition as research sites in hydrology and have been used since the late nineteenth and early twentieth centuries. The University of Natural Resources and Life Sciences Vienna (BOKU) recently extended its experimental research forest site 'Rosalia' with an area of 950 ha towards the creation of a full ecological-hydrological experimental watershed. The overall objective is to implement a multi-scale, multi-disciplinary observation system that facilitates the study of water, energy, and solute transport processes in the soil-plant-atmosphere continuum. This article describes the characteristics of the site and the monitoring network and its instrumentation installed since 2015, as well as the data sets. The network includes four discharge gauging stations and seven rain-gauges along with observation of air and water temperature, relative humidity, and electrical conductivity. In four profiles, soil water content and temperature are recorded at different depths. In addition, since 2018, nitrate, TOC and turbidity have been monitored at one gauging station. In 2019, a programme to collect isotopic data in precipitation and discharge was initiated. All data collected since 2015, including, in total, 56 high resolution time series (with 10-min sampling intervals), are provided to the scientific community on a publicly accessible repository. The datasets are available at https://doi.org/10.5281/zenodo.3997140 (Fürst et al., 2020).

## 1 Introduction

Environmentally oriented water management depends on understanding hydrological processes and their dominant controls at different spatial and temporal scales. To investigate hydrological processes and their complex interactions with the environment, long-term measurements from multi-disciplinary hydrological observatories are required (Schumann et al., 2010; Blöschl et al., 2016). As the earliest hydrological observatories, experimental watersheds have been used as far back as the late nineteenth and early twentieth centuries (USGS Reynolds Creek, (Seyfried et al., 2018)). Given these long-term datasets, changes in the hydrological cycle, such as those resulting from climate warming, can be investigated in these watersheds (Bogena et al., 2018).

In recent decades, there has been growing recognition that hydrology (and its related disciplines) cannot be treated in isolation. Rather, hydrological processes driven by meteorological conditions are also strongly controlled by complex feedback mechanisms with biotic and abiotic systems (Porporato and Rodriguez-Iturbe, 2002). Therefore, hydrological experimental watersheds have gradually transitioned into multi-disciplinary experimental watersheds. A prominent example for this is the 'Critical Zone Observatories' research project, which was initiated in 2007 by the U.S. National Science Foundation (Anderson et al., 2018).

Understanding processes based on research conducted at individual catchments is limited to the physio-geographic conditions at the particular location. In an effort to understand hydrological processes based on a wider spectrum of boundary conditions, networks of multi-disciplinary hydrological observatories have been established in recent decades. Examples of such networks

are the German 'TERrestrial ENvironmental Observatory network' (TERENO) (Zacharias et al., 2011), the 'International Network for Alpine Research Catchment Hydrology' (Bernhardt et al., 2015), the 'US National Science Foundation's National Ecological Observatory Network' (NEON) (Kampe et al., 2010), and the 'Euro-Mediterranean Network of Experimental and Representative Basins'(ERB) as part of UNESCO FRIEND (Flow Regimes from International Experimental and Network
Data) (Holzmann, 2018).

A prominent example of an observatories network is the 'Long Term Ecosystem Research' (LTER) initiative, which aims to better understand the structure and functioning of complex ecosystems and their long-term response to environmental, societal and economic pressures at different spatial scales (LTER Network Office, 2020). The LTER was initiated in 1980 with six US catchments and has since expanded to other continents, comprising different ecosystem types, climates, and pressures. The
LTER was further developed into the 'Long Term Socio-economic and Ecosystem Research' (LTSER) platform to emphasise the importance of the human dimension and to explicitly consider the socio-economic system in multi-disciplinary ecosystem research (Haberl et al., 2006). The European LTER is the 'European Long-Term Ecosystem, Critical Zone and Socio-Ecological Systems Research Infrastructure' (eLTER RI), which was established in 2003 by the European Commission as a 'European Strategy Forum on Research Infrastructures' (ESFRI) (ESFRI, 2020).

These networks of observatories make it possible to address some open research questions in hydrology that were recently formulated (Blöschl et al., 2019). The most challenging questions regarding catchment hydrology relate to the effect of small-scale variability on the upscaling of model parameters and processes, the transfer of model parameters to other (especially ungauged) catchments, and the derivation of flow paths and residence times of water and solutes in the subsurface at different scales. Overcoming these challenges requires the existing networks of observatories to be complemented in their
instrumentation and observational capacities, harmonising temporal and spatial frequencies and continuously monitoring natural tracers such as ions, metals and stable isotope ratios such as $^2H/^1H$, $^{18}O/^{16}O$ and $^{15}N/^{14}N$ in precipitation, discharge, and the subsurface. The Plynlimon research catchment in the U.K. (Neal et al., 2011; Cosby and Emmett, 2020) and the Krycklan catchment study in Sweden (Laudon et al., 2013) are good examples of such research catchments, with long term tracer and hydro-geochemical data available.

The University of Natural Resources and Life Sciences in Vienna (BOKU) has a long tradition and extensive experience in operating multi-disciplinary experimental sites. BOKU has been using these sites for research purposes, to monitor environmental changes and climate change impacts, to develop new monitoring techniques, and to train students in applied research. One of BOKU's sites is the experimental research forest 'Rosalia', with an area of 950 ha that was established in 1875 to facilitate research and education, mainly in forestry disciplines (Figure 1). Several forest dieback studies were
conducted in the 1980s. In 2013, a 222 ha watershed within the Rosalia forest site was established as an eco-hydrological experimental watershed, and this Rosalia watershed became part of the Austrian LTER-CWN (Research Infrastructure for Carbon, Water and Nitrogen) initiative.

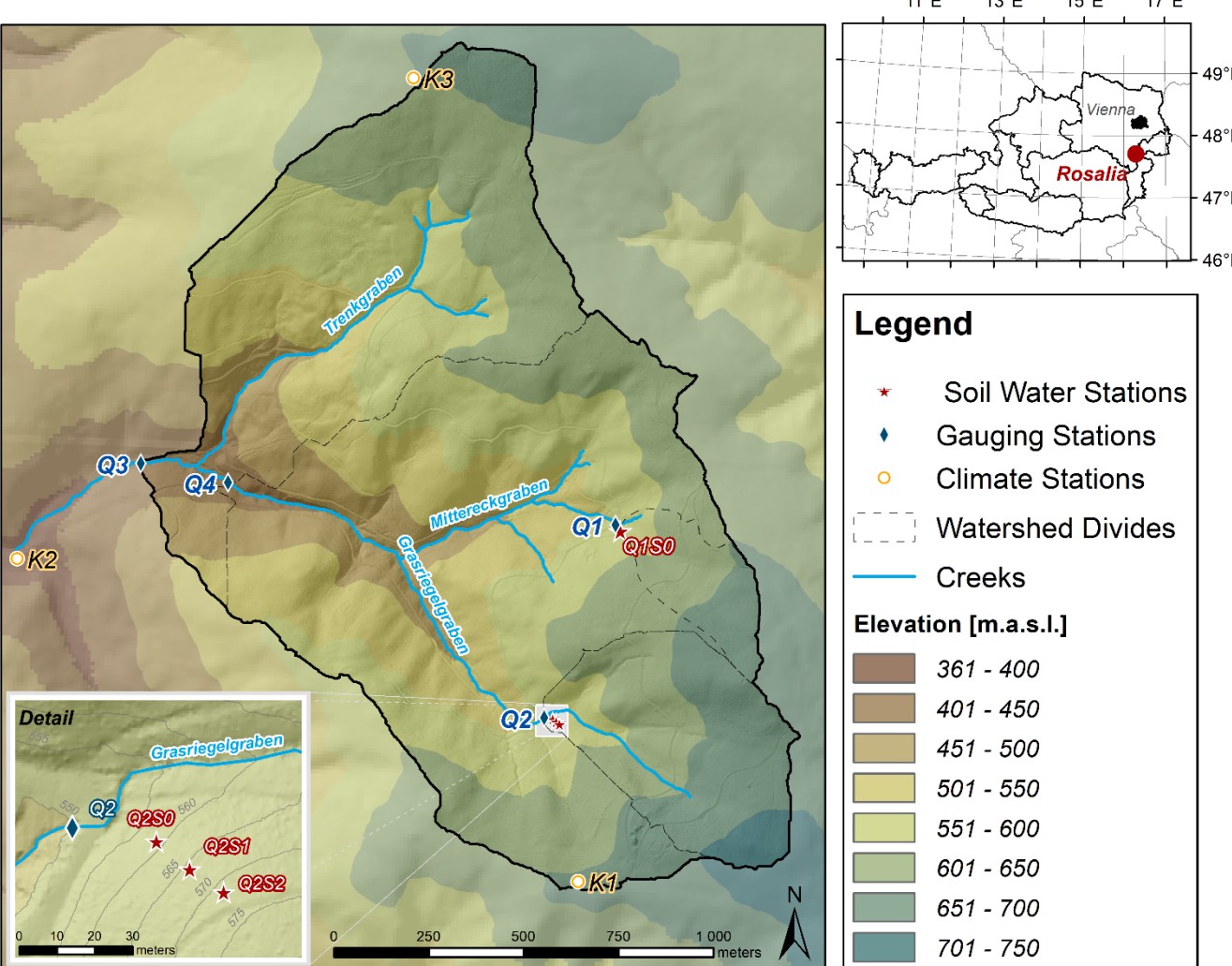

**Figure 1: Map of the watersheds and the monitoring network**

The overall objective is to implement a multi-scale, multi-disciplinary observatory that facilitates the study of water, energy, and solute transport processes in the soil-plant-atmosphere continuum. Research emphasis is put on deriving effective parameters for scales on which models simulate flow and transport processes (e.g. hillslope, catchment) by upscaling point measurements. A distinctive feature of the current monitoring setup is the continuous measurement of tracers in precipitation and discharge of selected creeks within the catchment, which allows deriving travel time distributions for sub-catchments and

investigating flow pathways in detail. Because BOKU has the right of access for educational and research purposes, large-scale controlled experiments can be undertaken. For example, rain-out shelters were used in parts of the forest by Netherer et al. (2015) to investigate drought impacts on bark beetle attacks on Norway spruce, while Schwen et al. (2015) and Leitner et al. (2017) used rain-out shelters to investigate soil water repellency and short-term organic nitrogen fluxes under a changing climate. Besides such local experiments, the monitoring network established since 2015 enables researchers to investigate the

impacts on the large-scale forest ecosystem and its services by providing the necessary baseline data. Investigating the transition of the forest ecosystem from its actual state into a pristine, unmanaged natural forest is among future research plans. The objective of this article is to present the monitoring network and the recorded data of the Rosalia watershed, and to make them available to the scientific community.

## 2 Description of the watershed

The Rosalia watershed is part of the Rosalia mountains (German: Rosaliengebirge) that belong to the eastern foothills of the Alps on the state border between Lower Austria and Burgenland in Austria (LAT 47°42'N, LON 16°17' E). Terrain heights

range from 320 to 725 m a.s.l., and the watershed is characterised by steep slopes (96% of the area is steeper than 10%, and 55% steeper than 30%). From 1989 to 2018, annual precipitation was between 560 and 1100 mm (average 790 mm, standard deviation 128 mm), and the mean annual air temperature was between 5.5 and 10 °C (average 8.2 °C, standard deviation 1.2 °C). Precipitation is not equally distributed throughout the year. Frequently in summer, heavy thunderstorms occur, causing floods that destroy forest roads, road culverts and other infrastructure (Figure 2).

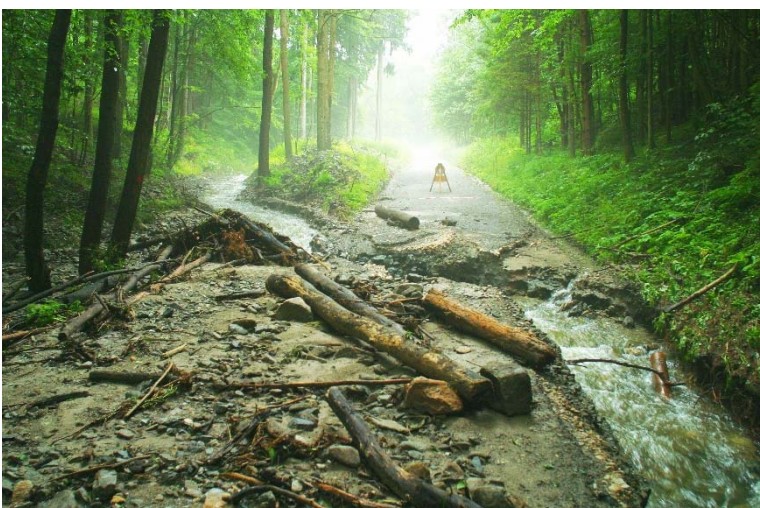

**Figure 2: Destructions of forest roads due to a storm on June 29, 2009 (Photo: J. Gasch)**

Crystalline rocks dominate in the Rosalia mountains, but coarse-grained gneiss, sericitic schist, phyllite and dolomite are also encountered. However, only coarse-grained gneiss with occasionally embedded dark or white mica schist are found in the actual catchment area of the hydrological research site.

The soils are predominantly cambisols that can be classified into four categories (Figure 3). The source materials for the recent soil formation are often remnants of tertiary soils that were modified by frost action and landslides during the ice age. The cambisols at steep slopes (slope >40%, category 1 in Figure 3) cover 5% of the area. They are podzolic cambisols with more than 40% coarse grain. These sites are characterised by poor water capacity and loss of organic material due to slope and wind. The characteristic species for these sites are beech with white woodrush (luzula albida) associated with pine (pinus sylvestris) and European larch (larix decidua) above 500 m a.s.l., while below 500 m a.s.l. they are associated with oak (quercus petraea). Cambisols at plains and moderate slopes (category 2 in Figure 3, 68% of the area) contain 30-50% coarse grain and have a medium water capacity. The characteristic species is beech with woodruff (galium odoratum). At higher elevations and cool north slopes, beech, spruce, and fir (abieti-fagetum) are found. Cambisol and planosol at plains and moderate slopes (category 3 in Figure 3, 22% of the area) are characterised by periodic water stagnation. They are typically on concave land forms and have good water capacity and nutrient sustenance. There is a risk of wind throw due to possible root dieback in long wet periods. Forest associations are the same as for category 2. Cambisol and fluvisol at valley slopes and bottom (category 4, 5% of the area) are characterised by varying contents of coarse material and profile thickness, but always have good nutrient and water supply. Where valleys form a flat bottom, fluvisols are the basis for plant growth. The dominant tree species on the slopes are ash and sycamore (aceri-fraxinetum), while ash and black alder (pruno-fraxinetum) dominate the valley floors.

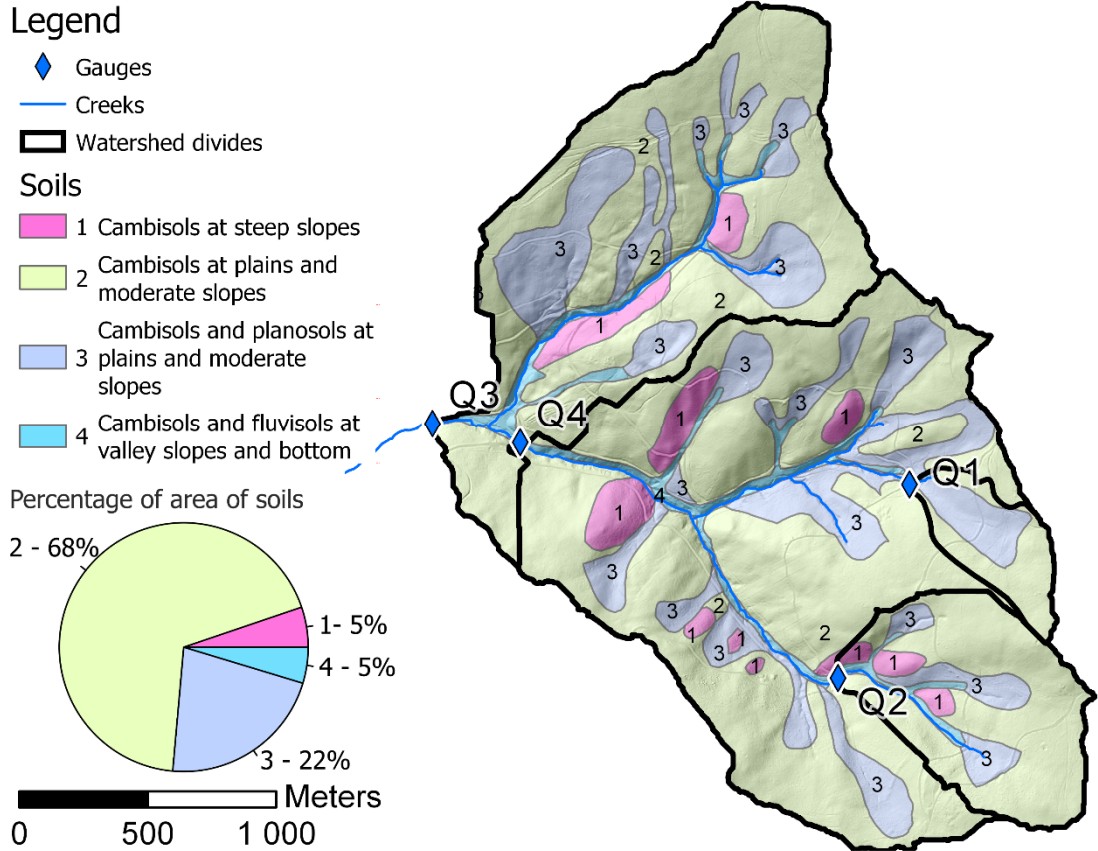

**Figure 3: Soil map with the four main soil categories, watershed divides and discharge gauges**

Forest management is performed by the Austrian Federal Forests (Österreichische Bundesforste, OeBf) owned by the Republic of Austria. BOKU has the right of access for educational and research purposes. OeBf manages the forest sustainably, balancing the protection of the environment, the needs of society, and economic success. The management of the forest is characterised by long production cycles of 100 to 140 years. The main species of the forest are the broadleaved beech (fagus sylvatica) and the coniferous Norway spruce (picea abies). The forest is at different development stages ranging from clear cut areas to mature forest stands. Natural regeneration is preferred to planting, and fertilisation is almost never done. Timber harvesting is usually done with harvesters and forwarders, and cable cranes are used at steep slopes. Management and timber transport are supported by a dense network of forest roads (50 m per hectare), suitable for heavy timber trucks. Main threats to the forest are snow break, wind throw and bark beetles, the latter affecting mainly coniferous tree species.

The main advantages of Rosalia as a research site are:

1. The watershed is part of the larger 950 ha forest site used by BOKU, and therefore a large amount of watershed information already exists, including soil maps, high-resolution DEMs (digital elevation model), maps on forest growth and productivity, detailed topographic maps, etc.

2. There is a well-established cooperation between BOKU and the owners of the forest, the Austrian Federal Forests, which facilitates even large-scale experiments with several years of duration.

3. Rosalia can be reached from Vienna within less than an hour, making maintenance cost-effective.

4. BOKU has an educational centre right at the border of the watershed, with seminar rooms, basic laboratory facilities and accommodation for up to 40 persons. Resident staff at the educational centre can assist in urgent situations, such as a storm or power failure.

## 3 Network of measurement sites

A network of stations (Figure 1, Table 1 and Table 2) has been set up to collect hydro-meteorological data: at four gauging
stations, river discharge, water and air temperature, relative humidity and electrical conductivity of water are monitored. The locations were selected to cover nested sub-watersheds of 9, 27, 146 and 222 ha, respectively. At one of these sites (Q4, 146 ha), water quality (NO$_3$-N, TOC, turbidity) is monitored with a spectrometer probe. Here, also stream water samples are taken for analysing stable isotopes of oxygen ($\delta^{18}$O) and hydrogen ($\delta^2$H). Precipitation is measured by seven rain gauges at different altitudes. At two of these locations, K1 and Q4, precipitation is additionally collected for the analysis of $\delta^{18}$O and $\delta^2$H. At four
locations, soil profiles were equipped with sensors measuring soil water content, electrical conductivity of soil water, and soil temperature at four and three depths, respectively.

**Table 1: List of sites, sensors and observed variables**

| Site | Sensors | Observed variables |
|---|---|---|
| **Q1**<br>**Mittereckgraben**<br>**559.87 m a.s.l**<br>**Watershed 9 ha** | Conductivity and temperature sensor Ponsel C4E | Electrical conductivity<br>Water temperature |
| | Rain gauge RG1 (Adcon tipping bucket) | 10-min rain depth |
| | Air temperature and humidity sensor TR1 (Adcon) | Air temperature<br>Relative humidity |
| | 1-ft H-Flume with 2 ultrasonic distance sensors (Baumer) | Water level in H-Flume<br>Discharge |
| | Tipping bucket, 1liter per tip (for discharge < 0.02 l/s) | Small discharge |
| **Q1S0**<br>**Soil water profile** | 4 HydraProbe soil sensors (Stevens) at Q1S0:<br>Sensor depths: 10, 20, 40, 60 cm | Soil water content<br>Soil temperature<br>Electrical conductivity of soil water |
| **Q2**<br>**Grasriegelgraben**<br>**550.06 m a.s.l**<br>**Watershed 27 ha** | Conductivity and temperature sensor Ponsel C4E | Electrical conductivity<br>Water temperature |
| | Rain gauge RG1 (tipping bucket) | 10-min rain depth |
| | Air temperature and humidity sensor TR1 | Air temperature<br>Relative humidity |
| | 1-ft H-Flume with 2 ultrasonic distance sensors | Water level in H-Flume<br>Discharge |
| **Q2S0**<br>**Soil water profile** | 4 HydraProbe soil sensors at Q2S0:<br>Sensor depths: 10, 20, 40, 60 cm | *Parameters see above* |
| **Q2S1**<br>**Q2S2**<br>**Soil water profiles** | 3 HydraProbe soil sensors at Q2S1 und Q2S2<br>Sensor depths: 10, 20, 40 cm | *Parameters see above* |
| **Q3**<br>**weir Grasriegelgraben**<br>**410 m a.s.l**<br>**Watershed 222 ha** | Depth sensor Keller PR46X | Water level at weir<br>Discharge |
| | Rain gauge RG1 (tipping bucket) | 10-min rain depth |
| | Air temperature and humidity sensor TR1 | Air temperature<br>Relative humidity |
| **Q4**<br>**Grasriegelgraben**<br>**415 m a.s.l**<br>**Watershed 146 ha** | 2-ft H-Flume with 2 ultrasonic distance sensors (Baumer) | Water level in H-Flume<br>Discharge |
| | Rain gauge RG1 (tipping bucket) | 10-min rain depth |
| | Air temperature and humidity sensor TR1 | Air temperature<br>Relative humidity |
| | S::can conductivity and temperature sensor condu:lyser | Electrical conductivity<br>Water temperature |
| | S::can multi::lyser spectrometer probe | TOC, NO$_3$-N, turbidity |
| | Palmex - rain sampler | Precipitation isotopes ($\delta^{18}$O, $\delta^2$H) |
| | Teledyne ISCO full-size portable sampler 6712 | River water isotopes ($\delta^{18}$O, $\delta^2$H) |
| **K1**<br>**Heuberg**<br>**640 m a.s.l** | OTT Pluvio² L – Weighing rain gauge | 10-min rain depth |
| | Air temperature and humidity sensor TR1 | Air temperature<br>Relative humidity |
| | Palmex - rain sampler | Precipitation isotopes ($\delta^{18}$O, $\delta^2$H) |
| **K2**<br>**Mehlbeerleiten**<br>**385 m a.s.l** | OTT Pluvio² L – Weighing rain gauge | 10-min rain depth |
| | Air temperature and humidity sensor TR1 | Air temperature<br>Relative humidity |
| **K3**<br>**Krieriegel**<br>**655 m a.s.l** | OTT Pluvio² L – Weighing rain gauge | 10-min rain depth |

**Table 2: Specifications of sensors**

| Sensor | Variable | Range | Resolution | Accuracy |
|---|---|---|---|---|
| **Adcon RG1 tipping bucket rain gauge** *http://www.adcon.at* | Precipitation [mm] | 0 – 200 mm/h | 0.2 mm | < 50 mm/h ± 1% 50 – 100 mm/h ± 3% 100 – 200 mm/h ± 5% |
| **Ott Pluvio² weighing rain gauge** *https://www.ott.com* | Precipitation [mm] | 12 – 1800 mm/h | 0.01 mm/min | ± 0.05 mm |
| **Adcon TR1 air temperature and humidity** *http://www.adcon.at* | Air temperature [°C] Relative humidity [% rH] | -40 - +60°C 0 - 100%rH | ± 0.1 °C | ± 0.1 °C ± 1% rH at 0 - 90% rH ± 2% rH at 90% - 100% rH |
| **UGT – 1 Ft- H flume** *https://www.ugt-online.de* | Discharge [l/s] | 0.02 – 55 l/s | | 2 – 5% |
| **UGT – 2 Ft- H flume** *https://www.ugt-online.de* | Discharge [l/s] | 0.04 – 315 l/s | | 2 – 5 % |
| **Keller PR-46X water level** *https://keller-druck.com* | Water level [m] | 0 – 1 m | < 1 mm | ± 0.55 mm |
| **Ponsel C4E water temperature and electrical conductivity** *https://en.aqualabo.fr* | Water temperature [°C] Electrical conductivity [µS/cm] | 0 – 50 °C 0 – 2000 µS/cm | 0.01°C < 0.1 µS/cm | ± 0.5 °C ± 1% of the full range |
| **s::can condu::lyser ᵀᴹ water temperature and electrical conductivity** *https://www.s-can.at* | Water temperature [°C] Electrical conductivity [µS/cm] | -20 – 130 °C 0 – 500 000 µS/cm | < 0.1 °C 1 µS/cm | not specified ± 1% of value |
| **Stevens Hydraprobe II** *https://www.stevenswater.com* | Soil water content [cm³ cm⁻³] Electrical conductivity [dS m⁻¹] Soil temperature [°C] | Dry to saturated 0 - 20 dS/m -10°C - +65°C | | ± 3% ± 2% or ± 0.2 dS m⁻¹ ± 0.6°C |


## 3.1 Data acquisition

Although the observed variables have different temporal characteristics, it was decided to record all *in-situ* measurements (except stable water isotope data) at synchronous 10-min intervals to simplify data storage and organisation. For this purpose, a UHF radio telemetry network (ADCON telemetry by OTT Hydromet GmbH) was implemented, enabling data acquisition,

storage, and management by a web-accessible database management system (DBMS). At each monitoring site, different sensors are connected to a remote telemetry unit (RTU). Within the network, several RTUs store and transmit data to a base station (located at the education centre building) and receive control commands from the base station. Apart from physically connecting the sensors to the RTU and providing power supply (solar or external), all setup, parameterisation, etc. is done remotely via a web interface to the base station.

The DBMS addVANTAGE Pro, which is connected to the base station via an internet link, is the main interface for administrators, regular users and the public. It is ADCON's universal data visualisation, processing and distribution platform. It is fully web-based, runs on a reliable PostgreSQL database engine, and is fully scalable from a single user version for five RTUs to servers with thousands of clients and thousands of RTUs. addVANTAGE Pro was configured to provide intuitive diagnostic displays of the measured hydro-meteorological variables as well as of hardware state and broadcasting parameters.

Pre-defined conditions, such as power-failure or exceedance of certain thresholds in the data, can trigger e-mail alerts to site administrators to enable timely remediation of issues, avoiding or reducing gaps in the records.

Stable water isotope data are not automatically uploaded to the DBMS but samples are collected on-site and picked up manually by university staff for analysis in the laboratory. Precipitation samples are collected bi-weekly with totalisators with plans to refine the sampling interval to daily, while streamflow samples are collected as daily grab samples using an autosampler.

## 3.2 Description of sites

### Discharge gauges

The sites for discharge measurements were selected to collect data for nested sub-catchments of different sizes. It was possible to find locations just at culverts of forest access roads, which has several advantages: (i) the sites are accessible by car, which is important for cost-effective maintenance; (ii) they have a defined sub-catchment outlet; and (iii) the H flume devices could be mounted directly on culverts, which meant that the road embankments could be used to fully capture even larger flows. H flume devices were selected to measure discharge as they cover a wide range of flow rates and most sediments are flushed through due to their horizontal bottom (Morgenschweis, 2010). For sites Q1 and Q2 with a watershed size of 9 and 27 ha, respectively, the 1-foot H flumes can measure discharge from 0.02 up to 55 l s$^{-1}$, where the upper limit corresponds to an approximately 5-year flood discharge (at the 27 ha site). Site Q4, with a watershed of 146 ha, is equipped with a 2-foot H flume (Figure 4). Water level at the H flumes is measured by pairs of ultrasonic distance sensors. One of these sensors measures the depth to the water level and the second measures a fixed reference distance. With the ratio of known reference distance to measured distance, the depth to water level is corrected for the dependence of the speed of sound on air temperature and relative humidity. Although H flumes are comparatively insensitive to sediment accumulation, we developed a compressed-air flushing system to keep the outflow section and the water level reference point free of sediments and debris. Site Q3 (222 ha) was already constructed in the 1980's, using a Thomson weir (Thomson, 1859). The water level at Q3 is measured by a capacitive pressure transmitter.

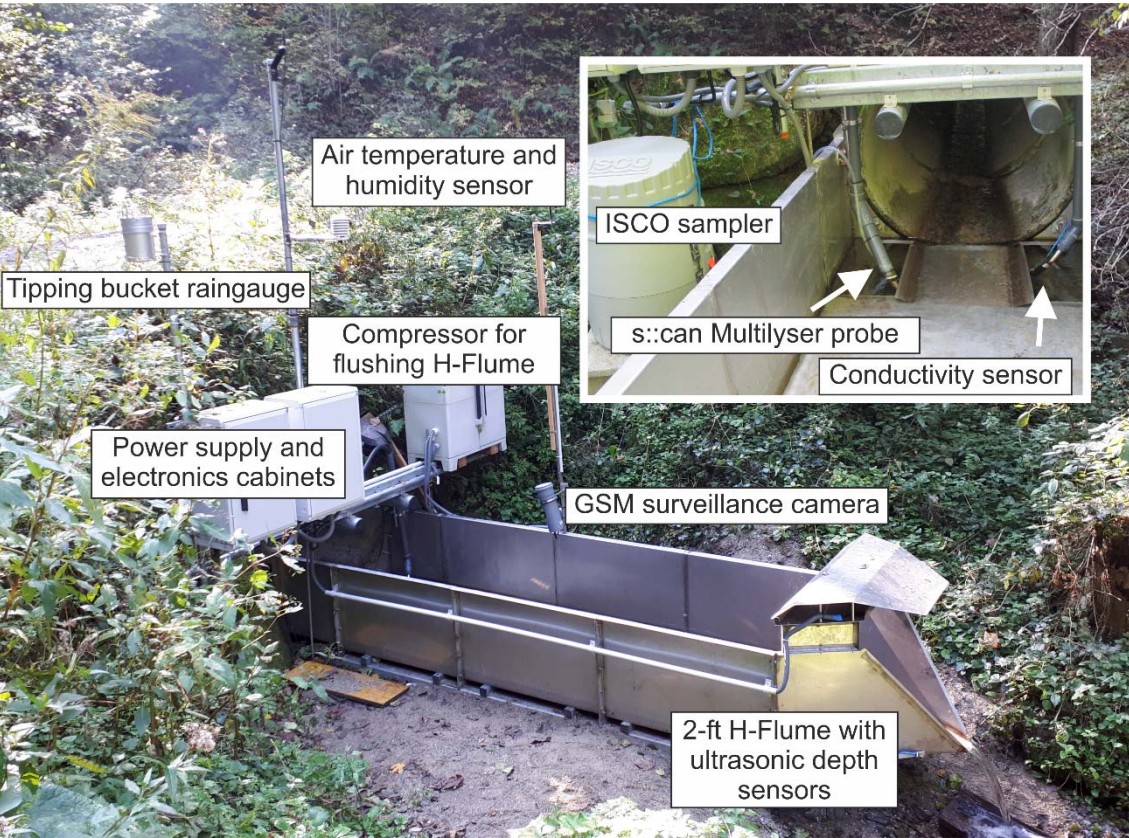

**Figure 4: Gauging site Q4 with 2-Ft H flume, spectrometer device and ISCO auto-sampler**

Sites Q1, Q2 and Q4 are additionally equipped with sensors for electrical conductivity, water temperature, air temperature and relative humidity. At sites Q1 and Q2, Ponsel C4E sensors (four electrodes) were installed to measure water temperature and conductivity, as they have an SDI-12 interface and low power consumption. They work electronically reliably, but the measured conductivities are sensitive to biofilms on the sensor, and the internal firmware requires more than an hour to achieve a stable reading after power-on or after cleaning. Furthermore, the measured conductivity tends to show an offset compared to manual measurements conducted approximately bi-weekly. Nevertheless, the recorded curves show plausible dynamics, e.g.,

during storm events. Currently, alternative sensors are tested to replace the C4E devices. At site Q4, a different type of sensor (s::can condu::lyser ™) is used, which, after more than a year of operation, recorded reliable and stable data.

**Rain gauges**

Sites K1, K2, and K3 are equipped with weighing rain gauges OTT Pluvio². Antifreeze fluid is added during the frost period, so that continuous measurements are possible. At the discharge sites Q1 to Q4, tipping bucket rain gauges are installed. They
require more maintenance than weighing rain gauges because the funnel is easily blocked by deposition of leaves, pollen, dust or insects, and they are inoperable during frost. Records from November to April must therefore be carefully checked using air temperature records and comparing the data with the records from the weighing rain gauges. Also, in the forest, it was not possible to follow all the rules for the proper placement of a rain gauge. Particularly, the recommendation that the height of nearby objects, such as trees, should not exceed the distance from the gauge to the objects (WMO, 2008), had to be disregarded
for Q1 and Q2. In particular, the rain gauge at Q1 is directly affected by the interception of the trees above. Two rain totalisators (Palmex Ltd., Croatia) were installed to collect precipitation samples for isotope analysis at meteorological station K1 and discharge site Q4.

**Soil water**

Stevens® HydraProbe® soil sensors (Stevens Water Monitoring Systems, Inc., Portland, OR, USA) were installed to
simultaneously measure soil moisture, temperature, and salinity (Stevens Water Monitoring Systems, 2015). The sensors deliver a standard data packet of six variables, including three variables characterising the dielectric properties of the soil and the resulting values of soil water content, temperature, and bulk electrical conductivity. The sensor-internal calculation of soil water content refers to the general calibration function published by Seyfried et al. (2005). In total, four soil profiles were equipped with HydraProbes. In two of the profiles, the sensors were installed at depths of 10, 20, 40, and 60 cm below the
surface (Figure 5); in the others, the sensors were installed at 10, 20, and 40 cm depth. Soil profile Q1S0 is located approximately 20 m upslope of gauge Q1. Soil profiles Q2S0, Q2S1, and Q2S2 form a transect up the slope line at 16, 30, and 45 m distance from Q2. This design supports a transect of soil water parameters measured along the slope line.

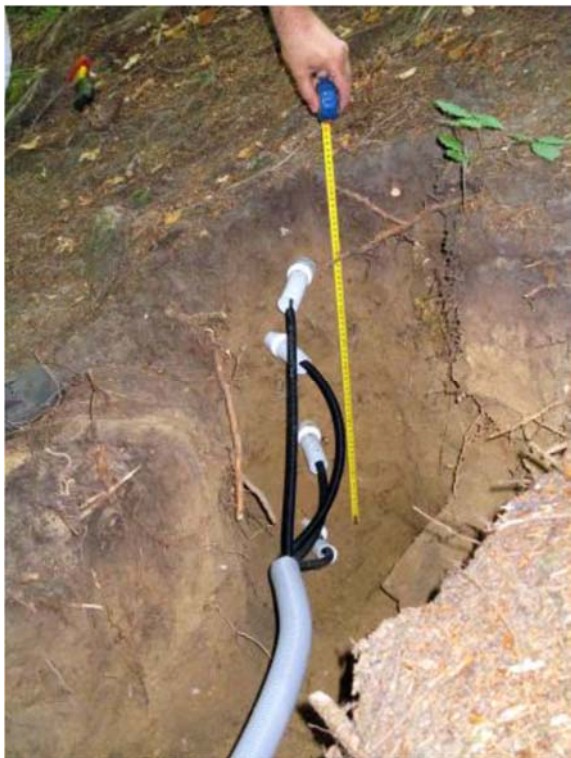

**Figure 5: HydraProbe sensors installed at site Q2S0**

**Water quality**

Since 2018, the water quality parameters $NO_3$-N, TOC, and turbidity have been monitored with a spectrometer probe s::can multi::lyser™ at site Q4. Starting in June 2019, a Teledyne ISCO full-size portable autosampler with a capacity of 24 1-litre bottles (model no: 6712) was installed at the same site to collect water samples for the laboratory analysis of $\delta^{18}O$ and $\delta^2H$. A daily sampling interval with 500 ml of water per sample was chosen to cover long-term changes in base flow and allow for daily snapshot information in case of events. The amount of water ensures a statistically sound sample size, while the sampling interval is short enough to enable the investigation of runoff events and is long enough that the autosampler can be left in the field for 24 days without maintenance. The suction tube leading from the H flume to the autosampler is occasionally affected by frost. The frozen water inside the tube prevents the autosampler pump from collecting water samples. Since the installation of the system, this has happened only rarely (less than 20 days), and we plan on further measures to mitigate freezing issues arising from small amounts of residual water in the tube that the pump cannot fully flush out. A potential evaporation issue arises from the fact that the autosampler is not a cooled field sampler. Hence, we manually collected streamflow grab samples each time the field site was visited and compared their measured isotope ratios to those of the sampling bottle which was standing the longest in the field. Preliminary results indicated no evaporation enrichment problem. Nonetheless, to minimise possible evaporation effects we are currently adapting the sampling bottles according to a recent publication (von Freyberg et al., 2020).

Close to the autosampler, open precipitation samples are collected approximately bi-weekly with a totalisator station (Palmex Ltd., Croatia) which is suitable for isotope sampling (Gröning et al., 2012). The sample bottle is inside a plastic pipe and thus protected from direct sunlight. The tube that connects the sample bottle to the funnel outlet has a small diameter and extends to the bottom of the sample bottle to limit air exchange. Since the collected rainfall at Q4 is not affected by interception, the samples did not undergo canopy-induced changes of the isotopic ratio that can influence the results of hydrologic models (Stockinger et al., 2015). Additionally, a Palmex totalisator station was installed at K1 to consider elevation effects on isotope ratios and sampled approximately bi-weekly until September 2020. After September 2020, the totalisator is emptied daily during work days (Monday to Friday) by staff of the BOKU education centre.

$\delta^{18}O$ and $\delta^2H$ are analysed using laser spectroscopy (Picarro L 2140-i, Picarro Inc., Santa Clara, CA, USA) in the isotope laboratory at BOKU. A calibration with laboratory reference material calibrated against the Vienna Standard Mean Ocean Water and Standard Light Antarctic Precipitation scale was used. All values are given in delta notation, and the precision of the instrument (1σ) was better than 0.1‰ and 0.5‰ for $\delta^{18}O$ and $\delta^2H$.

**4 Data**

All time series data are recorded, stored and routinely visualised using addVANTAGE Pro. For comprehensive analysis, data are regularly exported into the frequently used and freely available time series management system HEC DSS and the management software HEC DSSVue (Hydrologic Engineering Center, 2010). HEC DSSVue has powerful visualisation features and provides a convenient graphical editor for the time series. During editing, obvious artefacts such as spikes generated during maintenance, occasional obstructions of flumes during storms, and similar disturbances, are removed from the raw data. The data cleaning is specific to the variables and is therefore discussed in detail in the respective sections below. For even more flexible and automated processing, as well as for publication, the HEC DSS database was converted into a simple SQLite database (Hipp et al., 2019), which provides efficient and simple access from different software tools, including Python and R (Müller et al., 2018).

As the implementation of the instruments started in spring 2015, the earliest time series are from sites Q1 and Q2 and start in May 2015. Until September 2015, rain gauges K1 and K2, soil water profiles Q1S0 and Q2S0, and stream gauge Q3 were also added and delivering data. Soil water profiles Q2S1 and Q2S2 were added in April 2016, rain gauge K3, and stream gauge Q4

in summer 2018. For the majority of the data, more than four years of records are currently available (spring 2021). Out of the five years of records available at the time of publication, only the years 2018 and 2019 are presented in the graphs below to maintain readability.

## 4.1 Discharge data

Raw discharge data at the H flume gauges Q1, Q2, and Q4 needed careful inspection and editing. First, spikes in the hydrographs (one or two consecutive values significantly exceeding the value before and after the spike) were attributed to random events such as a leave under the ultrasonic depth sensor and were automatically replaced by linear interpolation. Next, visually detected implausible discharges were replaced by linear interpolation where reliably possible, or deleted otherwise. As an example, occasionally during very low flow, single leaves can temporarily (a few hours) get stuck at the narrow outlet

of the flume and cause the water level to rise a few millimetres. Such events are clearly visible as plateau-shaped parts of the hydrograph and can be safely replaced by linear interpolation. At these gauges, the measurements have never been disturbed by freezing.

At the weir Q3, two issues required editing: 1) during very low flow, leaves and grass can occasionally get stuck at the weir crest, causing the water level to rise. These events can be detected in the images transmitted daily by a surveillance camera

and visually in the hydrograph. Such artefacts are replaced by linear interpolation; 2) during longer frost periods, the stilling basin may be covered by ice and therefore the discharge is no longer described by the weir formula. These situations can be detected by visual inspection of the hydrograph and comparison with the temperature. These parts of the records have been deleted.

Discharge is characterised by its wide range of values (Table 3). At Q3 (watershed outlet with 222 ha), low flows in summer

and autumn are frequently less than $3\,l\,s^{-1}$, while peak flows of more than $500\,l\,s^{-1}$ have occurred twice since 2015. Specific discharge does not vary significantly between the four watersheds and typically ranges from 1 to $2\,l\,s^{-1}\,km^{-2}$ during low to medium flow and up to $30\,l\,s^{-1}\,km^{-2}$ during peak flows (calculated from daily means).

**Table 3: Statistics of discharge records and of missing data**

| Site | Time period | Min discharge $[l\,s^{-1}]$ | Max discharge $[l\,s^{-1}]$ | Mean discharge $[l\,s^{-1}]$ | Percent missing |
|------|-------------|-----------------|-----------------|------------------|-----------------|
| Q1 | 01JUN2015 - 31DEC2019 | 0.05 | 8.11 | 0.27 | 3.3 |
| Q2 | 01JUN2015 - 31DEC2019 | 0.24 | 12.64 | 0.81 | 0.9 |
| Q3 | 01SEP2015 - 31DEC2019 | 1.75 | 582.34 | 7.55 | 6.8 |
| Q4 | 01JUL2018 - 31DEC2019 | 1.35 | 309.68 | 4.23 | 1.1 |

In the hydrographs for the period 2018 to 2019 (Figure 6), increased base flows in spring and early summer are evident, as well as sharp peaks after rainfall events. The zoomed-in hydrographs for July/August 2018 (Figure 7) illustrate characteristic diurnal fluctuations of discharge during no-rain periods in the vegetation period (see section Applications for more details).

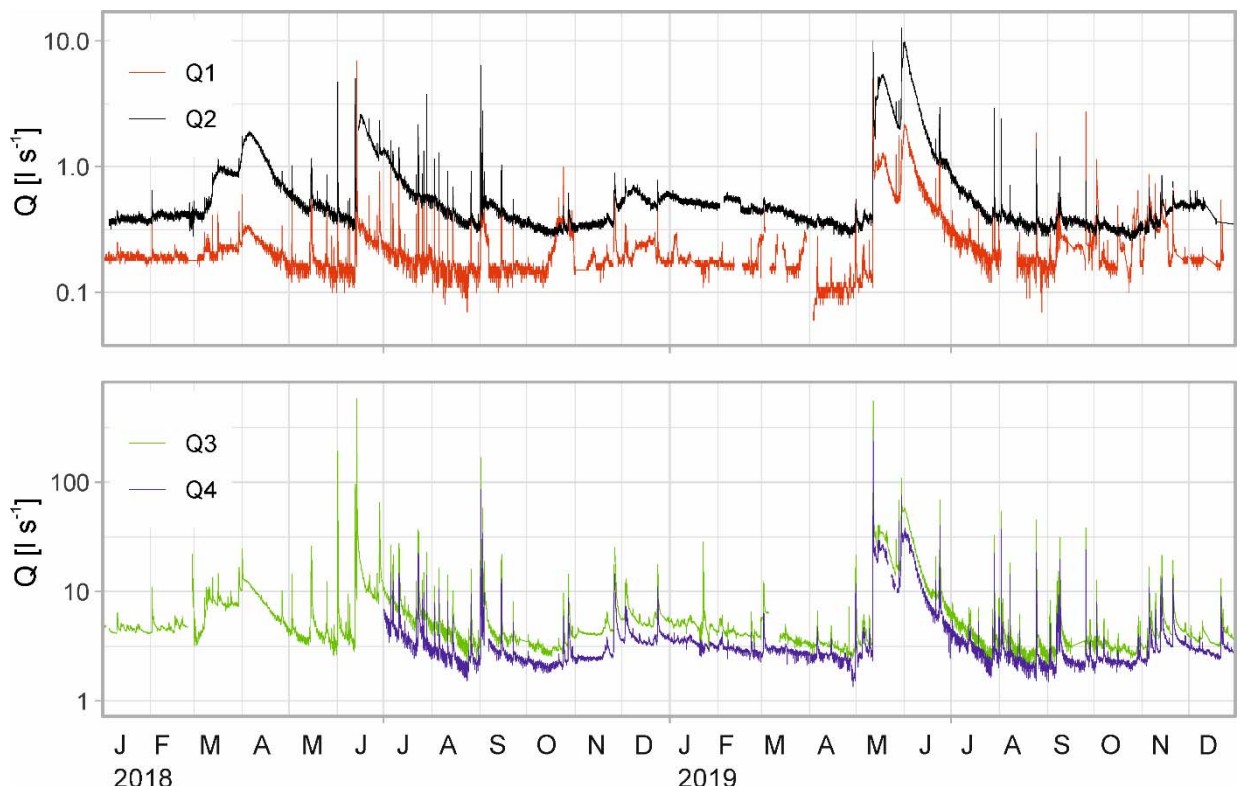

**Figure 6: Discharge hydrographs at gauges Q1 to Q4 for the years 2018-2019 (Q in log scale)**

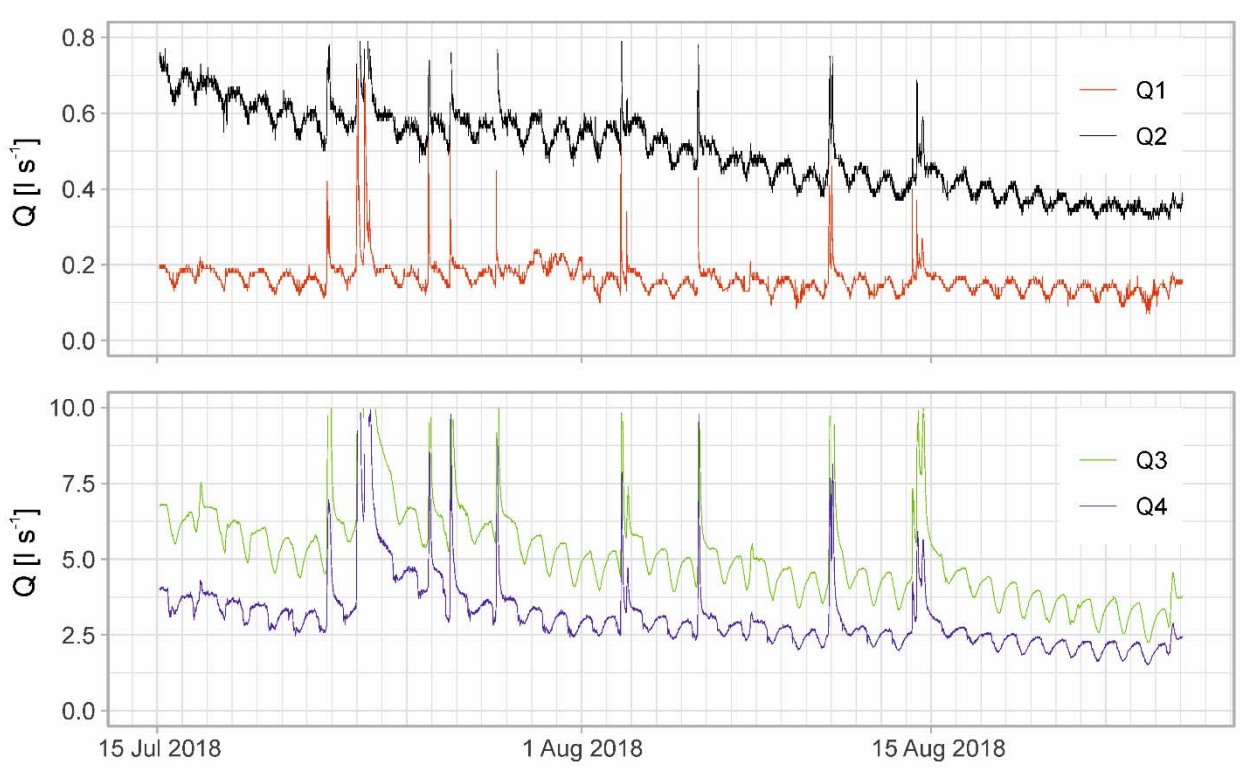


**Figure 7: Diurnal fluctuations of flow for July/August 2018 (peak flows are cut off: Q1, Q2 at 0.8 l s⁻¹, Q3, Q4 at 10 l s⁻¹)**

## 4.2 Precipitation data

For quality control, rainfall data recorded by tipping bucket devices (Q1 to Q4) are compared to records of the weighing rain gauges and to corresponding hydrographs. They are deleted if the funnel appears to have been (partially) blocked. Also, records

for the winter season from November to February are excluded due to tipping bucket issues with freezing. Anomalies observed during field maintenance visits (one to two per month) are also considered. The three weighing rain gauges have provided gap-

free records since the time of installation up to now, with a resolution of 0.1 mm. For most rainfall events between March and October, consistent and plausible data were acquired by up to seven rain gauges in total, providing a high-resolution rainfall pattern for a small area of 222 ha, and being spread over different altitudes from 385 to 655 m a.s.l (Table 4, Figure 8).

**Table 4: Statistics of precipitation data (statistics are calculated only if there are no missing values in the interval)**

| Site | Time period | Percent missing | Max daily precip. [mm] | Annual precipitation [mm] | | | |
|------|-------------|-----------------|------------------------|------|------|------|------|
| | | | | 2016 | 2017 | 2018 | 2019 |
| K1 | 26AUG2015 - 31DEC2019 | 0 | 69.2 | 975 | 676 | 877 | 759 |
| K2 | 26AUG2015 - 31DEC2019 | 0 | 60.8 | 949 | 682 | 906 | 739 |
| K3 | 01AUG2018 - 31DEC2019 | 0 | 84.1 | n.a. | n.a. | n.a. | 737 |
| Q1 | 01JUN2015 - 31DEC2019 | 26 | 56.6 | n.a. | n.a. | n.a. | n.a |
| Q2 | 01JUN2015 - 31DEC2019 | 29 | 63.0 | n.a. | n.a. | n.a. | n.a |
| Q3 | 01SEP2015 - 31DEC2019 | 31 | 48.6 | n.a. | n.a. | n.a. | n.a. |
| Q4 | 01JUL2018 - 31DEC2019 | 23 | 27.0 | n.a. | n.a. | n.a. | n.a |

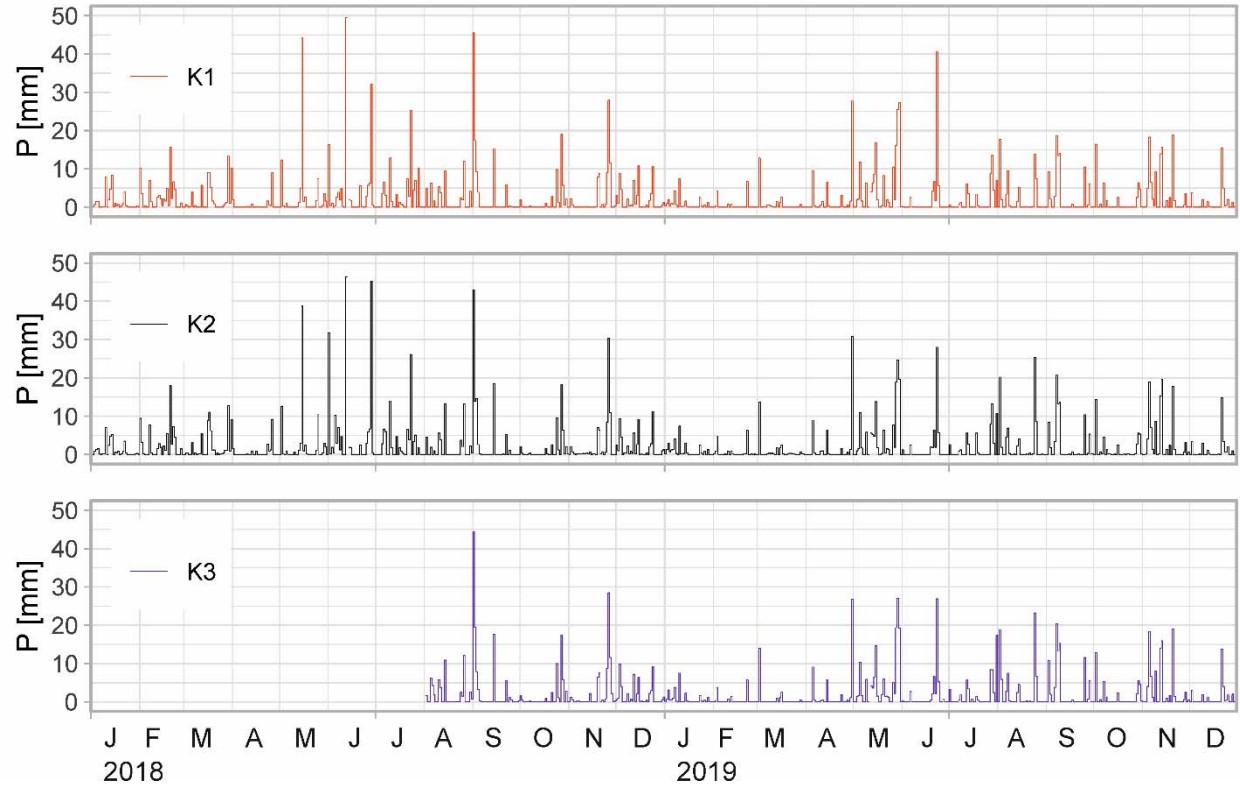

**Figure 8: Daily rainfall at the weighing rain gauges 2018 to 2019**

In this densely forested watershed, it is not possible to place all rain gauges at sites without interception or rain-shading.
However, a comparison of rainfall depths at all seven rain gauges for several events revealed good agreement. Gauge Q1 is affected by interception, which amounts to typically less than 2 mm per event (compared to weighing rain gauges K1 and K2), but monthly precipitation at Q1 is on average only 75% of the mean of K1 and K2. At Q2, monthly precipitation is on average 87% of the mean of K1 and K2. (K1 is close to the highest elevation of the watershed, K2 at the lowest – see Figure 1 and Table 1). Therefore, the data from all rain gauges are useful for analysing storm events, as interception reduces rainfall depths
by only a small percentage. For water balance investigations of periods longer than a week, however, only the gauges not affected by interception should be used.

**4.3 Soil water data**

With 14 HydraProbe sensors installed, and each measuring six variables, 84 soil water-related time series at 10 min resolution are recorded, resulting in a large volume of data. In the data repository, only soil water content (SWC) and soil temperature

are provided. Apart from an initial power supply problem at Q2S2, these sensors worked without any problem or data loss and required no maintenance. Figure 9 illustrates daily SWC in four depths at profile Q2S0, together with daily rainfall data. It is important to mention that the installation of the sensors requires digging a trench, which causes considerable local disturbance of the soil. Despite careful refilling, local infiltration paths could be influenced, and data do not necessarily reflect natural conditions for some time after installation. During the first few months after installation, for example, deeper probes reacted

faster to rainfall than those close to the surface (Figure 10). This can be attributed to artificial flow paths along the walls of the trench and the cables, or to effects arising from interrupted and destroyed natural macropores like wormholes. However, direct effects due to installation practically disappeared after the first season.

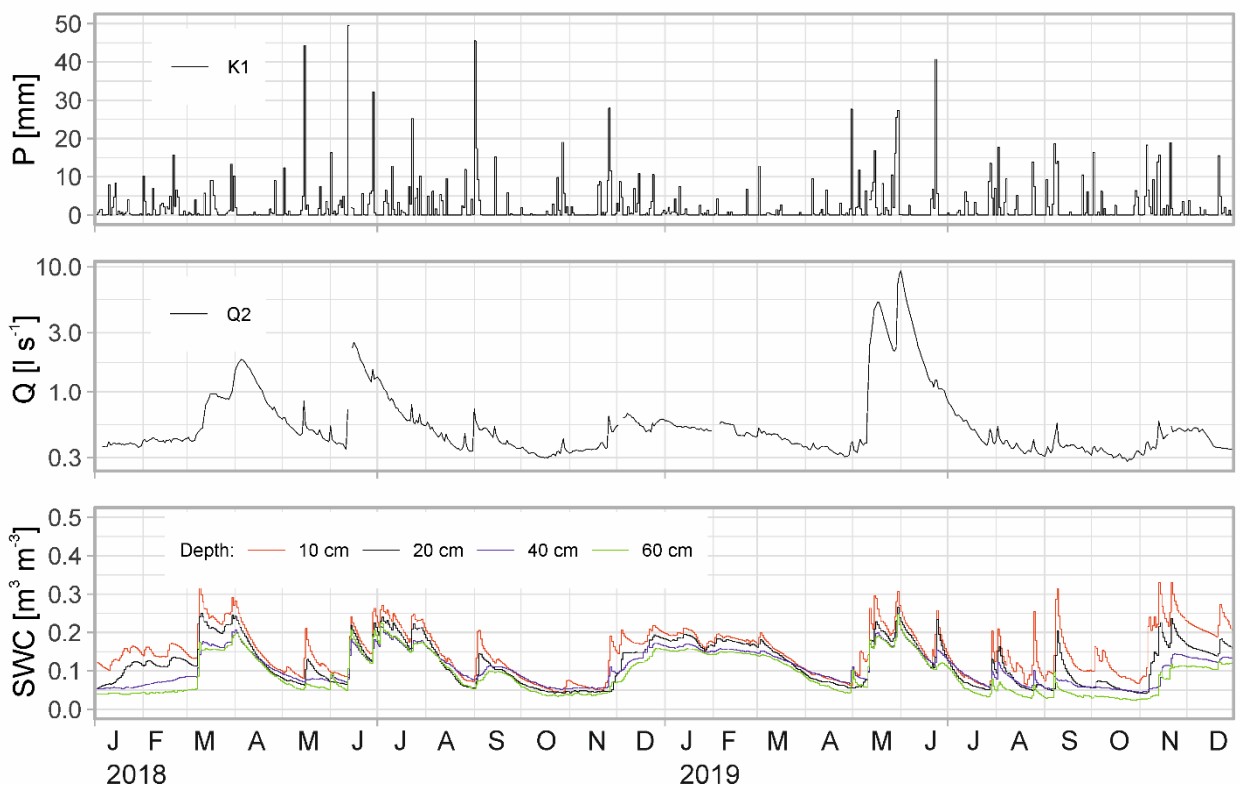

**Figure 9: Daily soil water content and corresponding daily rainfall and log-discharge at site Q2S0 for 2018 to 2019**

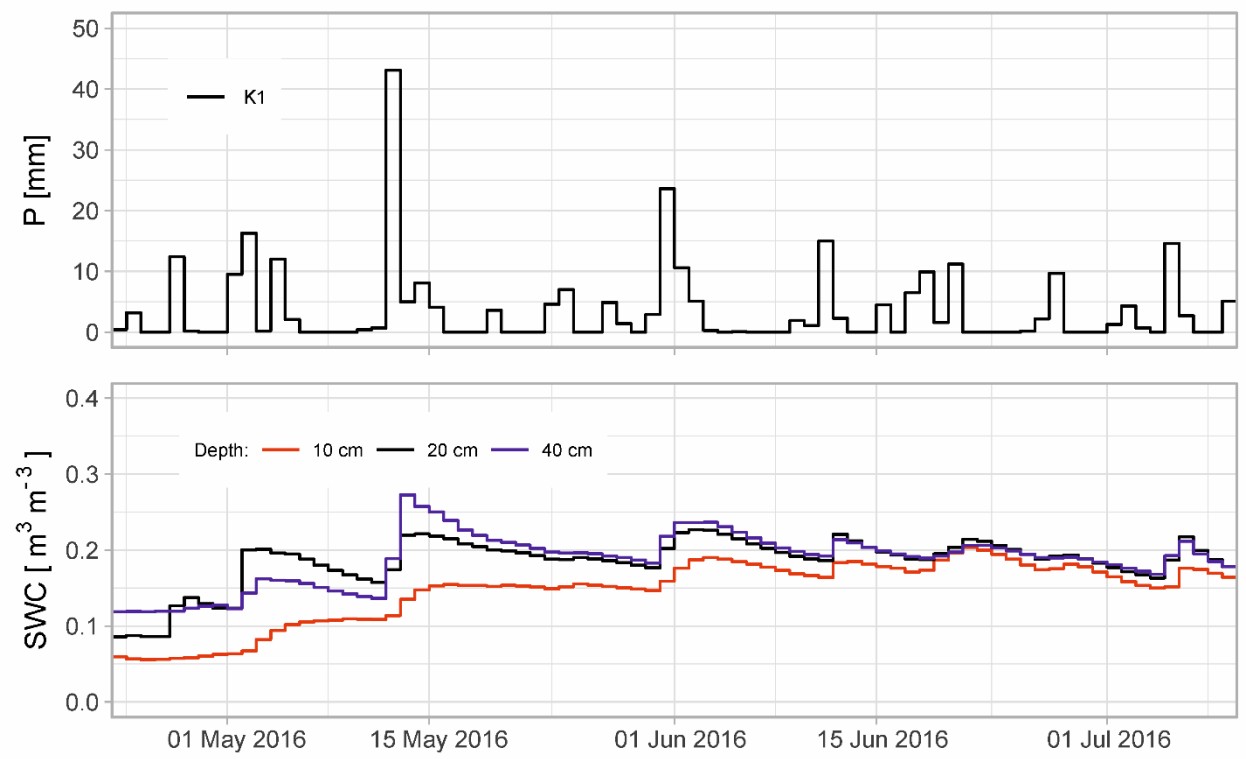


**Figure 10: Detail of daily soil water content at site Q2S1: deeper sensors reacted faster to rainfall on May 12, 2016**

## 4.4 Electrical conductivity and temperature of runoff

At discharge sites Q1, Q2, and Q4, water temperature and electrical conductivity are measured. Due to the risk of damage by frost, the sensors are removed during the frost period December to March at sites Q1 and Q2. Besides frost, conductivity
records at sites Q1 and Q2 are additionally negatively influenced by the sensor problems described in section 3.2. Regular conductivity measurements with a portable device showed that the conductivity of base flow is stable at sites Q1 and Q2 (typically approx. 120 µS/cm), so that the recorded conductivity series are still informative for the separation of base flow and direct runoff events, despite conductivity offsets in the records.

## 4.5 Isotopic data

At discharge site Q4, river and precipitation samples have been collected since June and October 2019, respectively (Figure 11). The precipitation data are collected as bi-weekly bulk samples and are compared to the daily river water grab samples. The comparison shows the response of the discharge to the precipitation input tracer signal (Figure 11). Furthermore, the precipitation and river water isotopes vary seasonally, with larger values in summer and lower values in winter months. This seasonality originated from contributions of precipitation to discharge, and isotope ratios in precipitation seasonally vary due
to changes in temperature, sources of vapour for cloud-formation and different rain-out histories (Feng et al., 2009). Apart from this, there are some preliminary indications of different flow paths, such as base flow (relatively stable $\delta^{18}O$ isotope values around -10 ‰), interflow (moderate increases or decreases in isotopes, for example at the beginning of August 2019), and faster flow (sharp peaks) suggesting dynamic runoff processes and transit times in the Rosalia watershed, which will be analysed in the future.

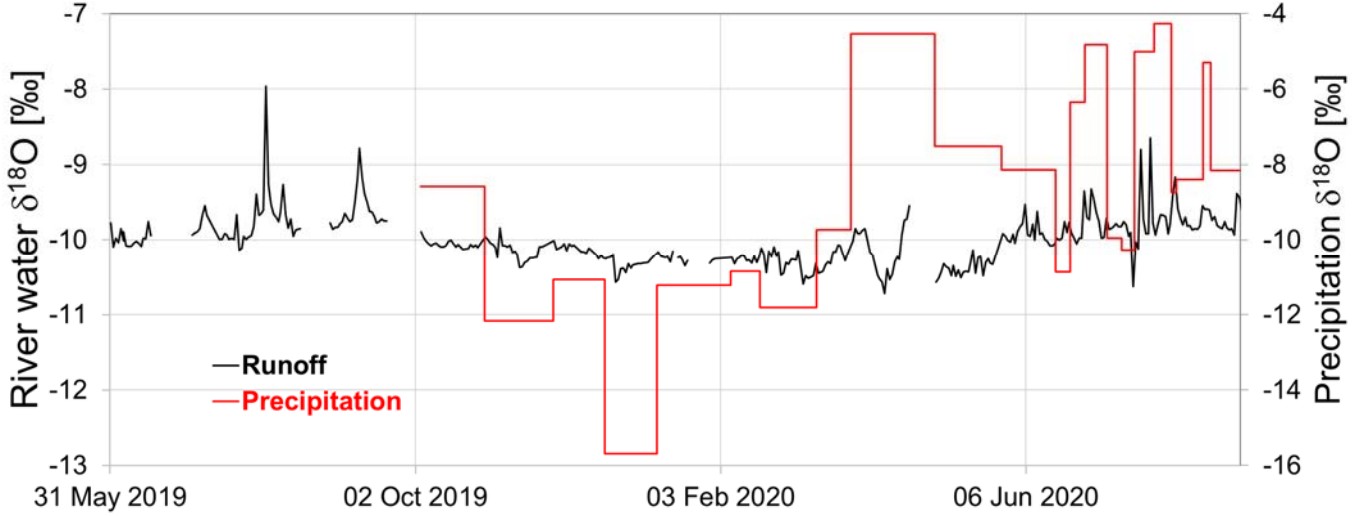

**Figure 11: Precipitation and river water δ¹⁸O isotopes at site Q4**

### 4.6 Spatial data

Data interpretation is complemented by a comprehensive amount of spatial data characterising the site. DEMs at various resolutions are available, including a 10 × 10 m DEM provided by the government of Austria, and a LIDAR based DEM at 0.5 × 0.5 m (Immitzer, 2009), accessible at https://zenodo.org/record/4601057. From these DEMs, watershed divides and the drainage network were derived in GIS. Additionally, a ground survey was performed for the main creeks in 2018. These data are included in the repository in shapefile format.

### 5 Applications

The presented data are suitable for studying processes of water flow and transport in small, forested watersheds. They have been used in academic teaching and research. The site is regularly used for advanced field courses in the water management and environmental engineering curriculum. During these courses, students not only learn about the setup and operation of a hydrological monitoring network, but they also contribute to the improvement of knowledge about the watershed by collecting and analysing soil samples or performing validation measurements of the instruments.

The dataset provided the majority of the database for two master's theses and a dissertation. Irsigler (2017) applied discharge and electrical conductivity data in a simple two-end-member mixing model for the separation of base flow and direct runoff, using an approach described by Lott and Stewart (2016). Stecher (2021) investigated a phenomenon that is observed in no-rain periods during the vegetation period: daily fluctuations of discharge, with peaks at 8 a.m. up to 40% higher than the minimum at 5 p.m., occur consistently at all four gauging sites. It was hypothesised that this is an effect of forest transpiration, since these diurnal fluctuations are not observed from late autumn to early spring. By modelling a slope transect at site Q2 with HYDRUS 2D (Simunek et al., 1999), the diurnal fluctuations of discharge are demonstrated to be caused by the vegetation in the riparian zone within only a few metres of the creek. Besides the discharge and rainfall records at site Q2, the model also used soil moisture data at sites Q2S0, Q2S1, and Q2S2.

Wesemann (2021) investigated the influence of forest roads and skid trails on runoff during heavy rainfall events in the Rosalia catchment. Based on the 0.5 × 0.5 m LIDAR DEM (Immitzer, 2009), he reconstructed a historical terrain model without forest roads and buildings, which allowed the comparison of the runoff from the natural terrain surface and runoff from the current surface, where flow paths are modified by the forest roads. The physically-based rainfall-runoff model RoGeR (Steinbrich et al., 2016) was set up for the catchment to quantify the influence of the road network on the runoff behaviour for three flood events, observed at gauge Q3 between 2017 and 2019. Rainfall data from all seven rain gauges were used to assess the effect of the spatio-temporal distribution of rainfall on runoff.

## 6 Data availability

All time series data were cleaned from the most obvious errors and artefacts and stored in an easily useable database. In addition, some auxiliary spatial datasets are made available. The data described above are available at https://doi.org/10.5281/zenodo.3997140 (Fürst et al., 2020). This repository comprises a SQLite database file with all the high-resolution time series data, an MS Excel sheet with the isotopic data, and the spatial datasets. Usage of the data is described by a comprehensive HTML file (generated by an also included R Markdown document), which includes previews and a full technical description of the data, including R code chunks to read and visualise them. The data repository will be updated annually.

## 7 Summary

The data represent an effort to measure components of the energy and water cycle in a forested catchment in the Eastern Austrian Alps. The period of record for some components started in 2015. Making the data available to the research and applied hydrology communities has two main objectives. First, it intends to inform decision-makers in the Rosalia forest. The record is an important source of baseline data that can be used to assess the effect of disturbances, such as clear-cuts and changing forestry on hydrological processes. Second, these data are provided to allow others to also investigate hydrological processes, medium-term patterns and potential changes in this type of watershed. Measurements use consistent methods to ensure comparability within the research catchment. The data have proven fit for the purpose of supporting hydrological and hydro-meteorological process research.

**Author contributions.**

J. Fürst was involved in field work to collect the data discussed here, including selection and installation of the instruments, processing, quality assurance and quality control. H. P. Nachtnebel and K. Schulz handled strategic decisions and funding. J. Gasch provided advice on site selection and provided some spatial data. R. Nolz handled the selection and installation of soil sensors. M. Stockinger and C. Stumpp set up the isotope measurement network and maintain it. All the authors contributed to writing the manuscript.

**Competing interests.**

The authors declare that they have no conflict of interest. Names of products and companies are only mentioned for better understanding and traceability; none of the authors is in a dependency to any of the mentioned companies

**Acknowledgements.**

Over the years, several people have contributed to the implementation of the Rosalia test site, the operation of the instruments, as well as data collection and deserve recognition. These include Wisam Almohamed, Matthias Bernhardt, Laurin Bonell, Reinhard Burgholzer, Roman Eque, Heinz Fassl, Martin Hackl, Mathew Herrnegger, Freddy Kratzert, Thomas Lehner, Martin Lichtblau, Johann Karner, Philipp Proksch, Andreas Schwen, Wolfgang Sokol, Gabriel Stecher, Johannes Wesemann.

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
