# Peer review of "Rosalia: An experimental research site to study hydrological processes in a forest catchment"

_Earth System Science Data, 2020_

## Referee Comment (RC1) · Anonymous Referee #1 · 22 Dec 2020

This data publication aims to describe the Rosalia experimental watershed, and introduce the data that is collected between 2015 and 2019. The authors give a very detailed description of all sensors and data storage application used, which I feel came at the expense of more information about the actual watershed and data. The geological background is summarized in one sentence only, and no geologic, vegetation or soil maps are shown, which are key if other researchers are to work with the data. No information is given about the process of data cleaning, or the analysis of the isotopic samples in the lab. Since one of the main aims of a data publication is that other people can work with the data after, I suggest that the article is adapted so that such crucial (!) information is described, and other researchers can also work with the data. Some timeseries of the actual data are shown (i.e., of discharge, soil moisture, rainfall, elec-

trical conductivity and the stable isotopic composition of discharge and rainfall), but the presentation of these is very minimal. Furthermore, for or one of the figures the axes were not correctly chosen (i.e., cutting off part of the data), and the figure captions cover only the bare minimum of information.

When reading this article, I stumbled over numerous grammar mistakes, wrong interpunction, colloquial language, use of the imperial system, and sentences that were clearly not formulated in correct English. I felt like I was doing the final reading before submitting, rather than a review. I was surprised that this was the case because from the abstract it sounded like the Rosalia catchment is the flagship of BOKU, and its documentation thus would deserve adequate attention. In addition to the language of this article, the structure also clearly needs more time and attention. Some definitions and topics are introduced but not fully discussed, and come back multiple times in the manuscript. This does not help the future reader of this article to find the information needed.

I apologize for my lack of in-depth comments to this article, but this article needs more time and attention before an in-depth review can be helpful. I suggest that the authors take this task serious and resubmit after careful re-structuring and rewriting. Documentation for a long-term research site (1875!) should be more comprehensive than this, and should for instance also include a background of the most important findings and the mechanistic understanding of how this watershed functions, in addition to the missing information with regard to data processing as mentioned above.

Detailed comments:

L21: remove additionally

L24: one site of how many sites? The discharge gauging stations?

L24: nitrate is capitalized where it should not be

L28: remove 'their'

L32: Global change impacts, such as climate warming? I don't see how climate itself is a global change impact.

L33-35: Although I somewhat agree, who realized this? reference needed

L35: experimental catchments? remove sites

L45: unclear which framework is referred to

L57-65: why is the LTER not introduced together with the other networks?

L74: if the object "was and still is" the word "is" is sufficient to indicate that

L76: if this is "a research emphasis" what are other important points?

L79: what are 'point related measurements'?point measurements?

L79-80: please rephrase this sentence to provide more clarity.

L83: how does the set-up allow for these experiments, in comparison to other sites?

L89: "are and will be investigated by a team of researchers" this sounds as if the team is already chosen, and cannot be adapted anymore. This is contrary to what I would expect is the aim of publishing this article, which is to promote other researchers to also use the data that is being published in this publication.

L89: same comment as with "was and still is" in L77

L95-98: since this is such a standard article lay-out, I would suggest that the others consider removing this description.

L100-101: this sentence is gramatically incorrect.

L102: 'is' steeper than

111-112: gramatically incorrect sentence

L129: the names of the watersheds, and their respective sizes, have not been introduced yet.

L30: monitored "with" a spectrometer probe

Line 131-135: every sentence starts on a new line.

L312: which altitudes?

L136-137: this sentence is redundant because this is mentioned in the figure and table captions.

L148: please specify what the "DMBS addVANTAGE Pro' is directly when first mentioning it.

L154: can the authors be more specific about the treatment samples after being collected by the totalisers or as grab samples? How are these samples stored in the samplers to ensure that the chemistry and isotope samples can both be analyzed adequately?

157: The field courses are organized by students? Or should this be "by students during field courses"

L158: which other (LiDAR-based) DEMS are available? and, LiDAR is commonly spelled with a lower-case 'i'.

L161: what is a "hydrological" site? A site at which hydrological measurements are being performed? in this case, the word 'hydrological' is redundant, given the sentence that follows.

L163: new line started where not needed.

L169: grammatically incorrect sentence. L170: grammatically incorrect sentence.

L168: please use the metric system.

L178: Reference missing for the "Thompson" weir.

L181: is their SDI-12 interface really important to mention in this article? And if so, be specific as to why the SDI-12 interface is preferred.

L181-184: Colloquial language. Please rewrite.

L183-184: please rewrite to make the sentence clearer.

L189: should be "are' possible.

L190: atmospheric deposition of what? Salts, leaves? please be specific.

L193: please rewrite to clarify the meaning of the sentence. Also, please quantify and be specific about how the rain measurements are affected, and why they are reliable in this data publication.

L212: d18O and d2H are already defined earlier in the manuscript. Please use the short-hand notation to make the text more concise, or refrain from defining the short-hand notations..

L230: 'using' addVantage Pro?' or does the program also assess the data? If so, please be specific about which protocols are used.

L234: can the authors be more specific about this data cleaning process?

L247: redundant to describe what Figure 5 illustrates, because this is mentioned in the caption. Please refer to the figure in the text itself.

L247: hydrographs 'for' July and August 2018

L273: could it not also be due to natural preferential flow paths? and if not, why not? and since in L274 the natural pref. flowpaths are mentioned, please be more specific about the limits to the period at which the disturbance affected the measurements.

L288: I would expect to find this sentence in an introduction, not in a 'results' section

L298: reference?

L300: which stable isotope? oxygen18 I assume?

L305: I think the spatial data can be introduced where the DEMS are introduced first, and don't need a separate section dedicated to them.

L311: please avoid one-sentence paragraphs at all times.

L314: what are the assumptions to this two end-member mixing model, and are these assumptions valid in the Rosalia catchment? What is the influence of soil water during rainfall events, and what is the EC signature of soil water vs. groundwater?

Section 5.1: please be more specific and actually quantify the results of your baseflow separation (don't forget to include uncertainties).

L320: please provide a reference for end-member splitting analysis.

L343: please give a measure of how well they match, NSE for instance.

L348: please be more specific about the data cleaning process. This is a very important part of the data collection and publication process, and is not mentioned at all in the manuscript.

Table 1: what does 0.2 mm 'events' or 0.1 mm 'events' mean? usually, 0.2 mm is the resolution of individual tips.

Table 1: Does the "tipping bucket device" have any other specification?

Table 1: please also mention the size (i.e., area in ha) of the different sites.

Figure 1: The cities on the inset map of Figure 1 are unreadable, and even the font size of the different sites in the main figure are a bit small. The legend nor caption describes what the green shading or crosshatching indicates. What is a "relais" in this context?

Figure 3: Please use the metric system.

Figure 5: y-axis is too low (Q2 peak cut off).

Figure 10: "stream water" or "river water" isotopes rather than river isotopes.

Figure 11: in its current form, Figure 11 does not add much to the article. The precipitation and discharge timeseries have already been shown in previous figures, and the results of the end-member mixing analyses are not shown.

Figure 12: is this specific discharge or absolute discharge?

[Figure]

---

## Referee Comment (RC2) · Anonymous Referee #2 · 4 Jan 2021

1. General comments

This data description paper depicts the Rosalia experimental research site in Austria. It introduces the forested watershed and its characteristics, the monitoring stations and hydrological equipment, the recorded data since 2015, and finally two example studies.

The manuscript describes the sensors and data storage applications in detail, but it should be presented in a more consistent and structured way. In addition, full documentation of the sites and accuracies would be desirable for the understanding of readers and potential users.

The two examples give an insight into two aspects of the studies. However, because they are not the main focus of the paper, the explanations and discussions can only be

very brief here.

The datasets are available in the specified data repository. Data collected at the described sites since 2015 are provided. It comprises a documentation of the dataset, GIS and time series data.

2. Specific comments

Right in the third line of the abstract, the operation of the study area since 1875 is mentioned. The reader looks forward to a long-term data series and analysis. However, he/she is then disillusioned relatively quickly that it is only about the data analysis since 2015. Many graphs even show only two years 2018-2019. I therefore recommend defusing the initially high expectations by moving the long-term aspect from the abstract to the introduction chapter.

In order to understand the multiplicity of sites, sensors and measurement data, a comprehensive listing and description is necessary. This is only done partially because the reader has to compile the information himself. The following appears to be in need of improvement:

a) Fig. 1 shows sites of 2018, but Tab. 1 shows the status of March 2020. Is the 2018 status up-to-date and does it correspond to the 2020 status?

b) Where is Q2S0 in Fig. 1?

c) The function of R1 Relais (Fig. 1) is not mentioned in the text - is it relevant for understanding?

d) In Tab. 1 there are the sites Q1-4, K1-3, Q2S1 and Q2S2, but Q1S0 and Q2S0 are missing.

e) Chapter 3 - L127-137 – is difficult to understand and to match with Tab. 1 and Fig. 1. It would be helpful to insert the site numbers/names here. Otherwise, one has to pick up everything from these lines and the table and the next chapters.

f) It would also be helpful to add the watershed sizes to Tab. 1. The same applies also to the depths of the four soil profiles, as these are assigned very unspecifically in L134-135 and L202-203. A column with the measurement interval and start date of the sensors used to measure each parameter could also be added to Table 1. To estimate data quality and sources of uncertainties and errors, further details about the sensors, such as sensor accuracy and operating range, should be provided with the data. Data gaps to show the proportion of no-data values could also be visualised in a graph.

g) L127 what is measured: river discharge or water level?

h) Chapter 2: It would also be helpful to list the characteristics of the four sub-basins in more detail: Is there heterogeneity in geology, soils and slopes? Is further information on soil important for understanding? What are the elevation ranges within the sub-basins, are there differences between the sub-basins? A map could help for visualisation. How is the forest managed (maintenance measures, use practices, fertilisation, sustainability, roads and infrastructures)?

Chapter 4.1: Is the specific discharge (L245-246) related to site Q3? What about the other sites? As this is a data description paper - add mean and range for all four gauges. The same for chapter 4.4.

Chapter 4.5: Which method was used for the isotope analyses in the lab?

Chapter 5.2: This is an interesting topic, but too complex for this kind of data description paper. Therefore, some assumptions and relationships are unproven, not supported by numbers or graphs (∼L325-336). Exact model performance remains unclear, statistical indicators are missing. Reference to other studies and a discussion are also not provided. Therefore, a separation between an overview presentation in this data description paper and a scientifically sound analysis in an original research article would certainly make more sense.

3. Technical corrections

Fig. 1: German-language city names (Wien, München. . .) appear in Fig. 1; the English names would be appropriate for this map in an English-language paper.

Fig. 7: Same scale length or axis layout as in Fig. 5 enables a better comparison.

L314: Add 'electrical' for electrical conductivity.

4. References

Chapter 4.6: What is the source of the DEMs? Add references.

Reference list: L418-419 Roadmap & Strategy Report on Research Infrastructures – cite as in the text as European Strategy Forum on Research Infrastructures, 2020 or ESFRI, 2020.

Missing references in reference list which can be found in the text: Cosby and Emmett, 2020; Gröning et al. 2012; Hydrologic Engineering Center, 2010; Hipp et al., 2019 Klaus and McDonnell, 2013; Müller et al., 2018; McGuire and McDonnell, 2006; Stevens, 2015.

5. Data repository

Regarding the file 'Isotope_ESSD.xlsx' in Table 'Q4DailyIso' in the data repository: Strange or missing values are marked and explained in the column 'Comment'. But gaps of several days are only marked by a line but not by an explanation, e.g. from 25.06.2019 to 03.07.2019, from 16.08.2019 to 28.08.2019, from 20.09.2019 to 04.10.2019, etc.

---

## Author Comment (AC1) · 16 Feb 2021

**Josef Fürst (on behalf of the co-authors) Feb 16, 2021**
*Replies are formatted in blue, while original referee's text is black.*

Dear anonymous reviewer,
thank you for your thorough review and your effort to help improve our contribution. I understand your concerns and agree with most of them. I am confident that we can resolve the issues that you pointed out. Please find detailed replies below.

This data publication aims to describe the Rosalia experimental watershed, and introduce the data that is collected between 2015 and 2019. The authors give a very detailed description of all sensors and data storage application used, which I feel came at the expense of more information about the actual watershed and data. The geological background is summarized in one sentence only, and no geologic, vegetation or soil maps are shown, which are key if other researchers are to work with the data.
R: We agree and will provide a soil map in the revised manuscript. The geological background is indeed very uniform which can and will be described in a few sentences. We will provide a more detailed description of the vegetation. Unfortunately, we cannot include the datasets in the data repository because of restricted copyrights for some of those datasets. Nevertheless, some of these are published on other platforms (e.g., LTER) and we will provide links.

No information is given about the process of data cleaning, or the analysis of the isotopic samples in the lab. Since one of the main aims of a data publication is that other people can work with the data after, I suggest that the article is adapted so that such crucial (!) information is described, and other researchers can also work with the data.
R: We will add more information about data cleaning (see reply to the comment on L234 below). Stable isotopes were analyzed using a laser spectroscope (Picarro L2140-i, cavity ring-down spectroscopy).

Some timeseries of the actual data are shown (i.e., of discharge, soil moisture, rainfall, electrical conductivity and the stable isotopic composition of discharge and rainfall), but the presentation of these is very minimal. Furthermore, for or one of the figures the axes were not correctly chosen (i.e., cutting off part of the data), and the figure captions cover only the bare minimum of information.
R: The data repository contains comprehensive documentation and visualization of the time series data in an easily accessible and interactive HTML format. Therefore, we selected only 2 years for the figures in the paper to improve readability. We will add a note to the captions. The characterization of the time series will be extended in the text. The wrong axis scaling of Fig. 5 will be corrected in the revised version!

When reading this article, I stumbled over numerous grammar mistakes, wrong interpunction, colloquial language, use of the imperial system, and sentences that were clearly not formulated in correct English. I felt like I was doing the final reading before submitting, rather than a review.
R: We will pay more attention to a correct and consistent language and style in the revised manuscript. After revision the manuscript will undergo a proof-read by a native English speaker.
Regarding use of the imperial system: this is used only in the naming of the H Flume devices where it is, in our opinion, appropriate. H Flumes are standardized, off-the-shelf devices which are characterized by their depth in ft. Even European vendors and German textbooks on hydrometry (E.g. (Morgenschweis, 2010)) list H Flumes by their depth in ft (and not m). The rating curves for H Flumes have been developed by the US SCS for a range of different sizes and therefore it would not help any reader to write about a 0.305 m or 0.61 m H Flume.

I was surprised that this was the case because from the abstract it sounded like the Rosalia catchment is the flagship of BOKU, and its

documentation thus would deserve adequate attention. In addition to the language of this article, the structure also clearly needs more time and attention. Some definitions and topics are introduced but not fully discussed, and come back multiple times in the manuscript. This does not help the future reader of this article to find the information needed.

I apologize for my lack of in-depth comments to this article, but this article needs more time and attention before an in-depth review can be helpful. I suggest that the authors take this task serious and resubmit after careful re-structuring and rewriting. Documentation for a long-term research site (1875!) should be more comprehensive than this, and should for instance also include a background of the most important findings and the mechanistic understanding of how this watershed functions, in addition to the missing information with regard to data processing as mentioned above.

R: Mentioning that BOKU is using the Rosalia forest since 1875 was not meant to implicitly indicate that data are available for the entire time period. In fact, before we started to extend the site into a full eco-hydrological experimental watershed, it was predominantly used as an educational forest. To avoid any misunderstanding, we will re-structure the text to clearly focus on the monitoring network established and the data recorded since 2015.

Detailed comments:
L21: remove additionally
R: will be removed

L24: one site of how many sites? The discharge gauging stations?
R: at one new discharge gauging station (Q4)

L24: nitrate is capitalized where it should not be
R: will be corrected

L28: remove 'their'
R: will be corrected

L32: Global change impacts, such as climate warming? I don't see how climate itself is a global change impact.
R: will be re-phrased ("climate change")

L33-35: Although I somewhat agree, who realized this? reference needed
R: The importance of vegetation-hydrology interactions is highlighted in Porporato, A. and I. Rodriguez-Iturbe (2002). "Ecohydrology - a challenging multidisciplinary research perspective." Hydrological Sciences Journal-Journal Des Sciences Hydrologiques 47(5): 811-821. The reference will be added.

L35: experimental catchments? remove sites
R: will be corrected

L45: unclear which framework is referred to
R: will be re-phrased: "A recent report on the status …"

L57-65: why is the LTER not introduced together with the other networks?
R: we agree. LTER will be introduced together with the other networks

L74: if the object "was and still is" the word "is" is sufficient to indicate that
R: will be re-phrased: "The overall objective is to implement …"

L76: if this is "a research emphasis" what are other important points?
L79: what are 'point related measurements'?point measurements?
R: point measurements

L76-80: please rephrase this sentence to provide more clarity.
R: will be re-phrased: "Research emphasis is put on deriving effective parameters for scales on which models simulate flow and transport processes (e.g., hillslope, catchment) by upscaling point measurements."

L83: how does the set-up allow for these experiments, in comparison to other sites?

R: BOKU has management options in the forest that are generally not available in private forests where often just the implementation of monitoring stations is possible.
We will incorporate the following information into the revised manuscript: The forest management is performed by the Federal Forests of Austria (Österreichische Bundesforste, OeBf) which is owned by the Republic of Austria. BOKU has the right of access for educational and research purposes. OeBf claims to manage the forest according to sustainability principles, balancing protection of environment, the needs of society and commercial success. Management of the forest is characterized by long production cycles of 100 to 140 years. The main species are the broadleaved beech (fagus sylvatica) and coniferous Norway spruce (picea abies). Natural regeneration is preferred to planting. Fertilisation almost never occurs. Timber harvesting is usually done by means of harvesters and forwarders, at steep slopes cable cranes are used. Management and timber transport are supported by a dense network of forest roads (50m per hectar), suitable for heavy timber trucks. Main threats are snow break, wind throw and bark beetles, the latter affecting mainly coniferous tree species.

L89: "are and will be investigated by a team of researchers" this sounds as if the team is already chosen, and cannot be adapted anymore. This is contrary to what I would expect is the aim of publishing this article, which is to promote other researchers to also use the data that is being published in this publication.
R: With publishing the dataset we are certainly inviting other researchers to use the data too. We will rephrase our statement. "… strategies will be investigated by multidisciplinary teams of researchers".

L89: same comment as with "was and still is" in L77
R: will be re-phrased, see above.

L95-98: since this is such a standard article lay-out, I would suggest that the others consider removing this description.
R: will be removed

L100-101: this sentence is gramatically incorrect.
R: rather 101-103? Will be corrected: "The research watershed has terrain heights from 320 to 725 m asl, and it is characterized by steep slopes (96 percent of the area is steeper …"

L102: 'is' steeper than
R: will be corrected, see above

L111-112: gramatically incorrect sentence
R: will be corrected

**R: The description of the watershed (section 2) will be extended to provide more information on geology, soils and vegetation (see also reply to first comment above).**

L129: the names of the watersheds, and their respective sizes, have not been introduced yet.
R: we will improve this part and redesign Fig. 1 and Table 1.

L30: monitored "with" a spectrometer probe
R: will be corrected

Line 131-135: every sentence starts on a new line.
R: will be corrected

L312(132?): which altitudes?
R: altitudes will be added to table 1.

L136-137: this sentence is redundant because this is mentioned in the figure and table captions.
R: we will avoid such redundancies

L148: please specify what the "DMBS addVANTAGE Pro' is directly when first mentioning it.
R: short description of addVANTAGE Pro will be added.

L154: can the authors be more specific about the treatment samples after being collected by the totalisers or as grab samples? How are these samples stored in the

samplers to ensure that the chemistry and isotope samples can both be analyzed adequately?

R: Regarding precipitation samples, the totalisators are designed in a way to prevent isotopic enrichment by limiting evaporation (Groening et al. 2012 ). The sample bottle is inside a plastic pipe and thus protected from direct sunlight. The tube that connects the sample bottle to the funnel outlet has a small diameter and extends to the bottom of the sample bottle to limit air exchange. For streamflow samples, the ISCO automatic sampler is not a cooled field sampler. Since with 24 sample bottles and daily sampling intervals the first bottle is at maximum 24 days under ambient temperature conditions, there are concerns about evaporation enrichment. For this reason, we manually collect streamflow grab samples each time we visit the field site and compare their measured isotopes to those of the sampling bottle with the longest standing time. Although occasional deviations larger than 0.1‰ d18O occurred, preliminary results especially during summer months indicate no evaporation enrichment problem. Still, we'll adapt the system in future according to a very recent publication (van Freyberg et al. 2020 ).

157: The field courses are organized by students? Or should this be "by students
during field courses

R: "by" students during field courses

L158: which other (LiDAR-based) DEMS are available? and, LiDAR is commonly
spelled with a lower-case 'i'

R:  we will explicitly list all available DEMs. The acronyms Lidar, lidar, LIDAR, LiDAR, and LADAR all mean the same and are in use.

L161: what is a "hydrological" site? A site at which hydrological measurements are
being performed? in this case, the word 'hydrological' is redundant, given the sentence
that follows.

R: will delete "hydrological"

L163: new line started where not needed.

R: will be corrected

L169: grammatically incorrect sentence. L170: grammatically incorrect sentence

R: we will let a native speaker check the grammar

L168: please use the metric system.

R: Foot is appropriate in this context, see note above

L178: Reference missing for the "Thompson" weir.

R: Yes, it should read "Thomson" weir, which is a "90° V notch sharp crested weir". No primary reference was found so far. We can include a reference to a standard textbook on hydrometry.

L181: is their SDI-12 interface really important to mention in this article? And if so, be
specific as to why the SDI-12 interface is preferred.

R: it is not directly relevant for the dataset, but readers might be interested in the sensors that we use. SDI-12 sensors are generally known for their very low power requirements and their standard interface to most data-loggers.

L181-184: Colloquial language. Please rewrite.

R: will be re-phrased

L183-184: please rewrite to make the sentence clearer.

R: will be re-phrased. Measured conductivity tends to show an offset. Nevertheless, the recorded curves show plausible dynamics, e.g., during storm events.

L189: should be "are' possible.

R: will be corrected: … measurements are …

L190: atmospheric deposition of what? Salts, leaves? please be specific.

R: will be adapted: " … blocked by deposition of leaves, pollen, dust or insects, …"

L193: please rewrite to clarify the meaning of the sentence. Also, please quantify and
be specific about how the rain measurements are affected, and why they are reliable
in this data publication.

R: Specifically, the recommendation that the height of near-by objects, such as trees, should not exceed the distance from the gauge to the objects (WMO, 2008), could not be met at Q1 and Q2. A comparison of rainfall depths at all 7 rain gauges for several events revealed good agreement (*will be quantified in the revised manuscript*), however. Gauge Q1 is affected by interception, which amounts to typically less than 2 mm per event (compared to weighing rain gauges K1 and K2), but monthly precipitation is on average only 75 % of the mean of K1 and K2. At Q2, monthly precipitation is on average 87 % of the mean of K1 and K2. (K1 is close to the highest point of the watershed, K2 at the lowest point – this will be clear after re-design of Fig. 1 and Table 1). Therefore, the data from all rain gauges are useful for the analysis of storm events, where the interception reduces rainfall depth only by a small percentage. For water balance investigations of time periods longer than a week, however, only the gauges not affected by interception should be used. We will include this information in the revised manuscript.

L212: d18O and d2H are already defined earlier in the manuscript. Please use the short-hand notation to make the text more concise, or refrain from defining the shorthand notations.
R: will be corrected

L230: 'using' addVantage Pro?' or does the program also assess the data? If so, please be specific about which protocols are used.
R: for the purpose of this paper, "using" is sufficient.

L234: can the authors be more specific about this data cleaning process?
R: we will provide more information: Cleaning of raw data is done based on visual inspection of the raw data hydrographs. Also anomalies observed during field maintenance visits (1-2 per month) are incorporated.
**Rainfall data** recorded by tipping bucket devices (Q1, Q2, Q3, Q4) are deleted, if the funnel appeared to be (partially) blocked. Also, records for the winter season from November to February are excluded. Weighing rain gauges K1-K3 delivered continuous data since their installation.
**Discharge** at the H Flume gauges Q1, Q2 and Q4 needed careful inspection and editing. First, spikes in the hydrographs (1 or 2 consecutive values significantly higher than the value before and after the spike) were attributed to random events like a leave under the ultrasonic depth sensor and were automatically replaced by linear interpolation. Next, visually detected implausible discharges were edited, replaced by linear interpolation where reliably possible and deleted otherwise. As an example, occasionally during very low flow, single leaves can temporarily (a few hours) get stuck at the narrow outlet of the flume and cause the water level to rise a few millimeters. Such events are clearly visible as plateau-shaped parts of the hydrograph and can be safely replaced by linear interpolation. At these gauges, measurements were so far never disturbed by freezing.
At the weir Q3, two reasons required editing: 1) during very low flow, leaves and grass can occasionally get stuck at the weir crest, causing the water level to rise. Such events can be detected in the images transmitted daily by a surveillance camera and visually in the hydrograph. Such artefacts are replaced by linear interpolation. 2) during longer frost periods, the stilling basin may be covered by ice and therefore the discharge is no longer described by the weir formula. Such situations can be detected by visual inspection of the hydrograph and comparison with the records of temperature. These parts of the records are deleted.
The largest gaps are in the conductivity data at Q1 and Q2 (see separate discussion). Air and water temperature, relative humidity and soil water content did not need any editing and are practically gap-free.

L247: redundant to describe what Figure 5 illustrates, because this is mentioned in the caption. Please refer to the figure in the text itself.
R: will be adapted

L247: hydrographs 'for' July and August 2018
R: will be corrected

L273: could it not also be due to natural preferential flow paths? and if not, why not? and since in L274 the natural pref. flowpaths are mentioned, please be more specific about the limits to the period at which the disturbance affected the measurements.
R: During the first few months after installation, for example, deeper probes reacted faster to rainfall than those close to the surface (Figure 9). This can be attributed to artificial flow paths along the walls of the trench and the cables, or to effects arising from interrupted and destroyed natural macropores like wormholes. However, direct effects due to installation practically disappeared after the first season.

L288: I would expect to find this sentence in an introduction, not in a 'results' section

R: we will provide this information in the introduction and delete from the results section

L298: reference?
R: Feng et al. (2009 ): "Seasonality of isotopes in precipitation: A global perspective"

L300: which stable isotope? oxygen18 I assume?
R: yes, $\delta^{18}$O. will be added to the sentence

L305: I think the spatial data can be introduced where the DEMS are introduced first, and don't need a separate section dedicated to them.
R: it is more than just DEM, also watershed divides, surveyed creeks and location of sites. We will add a more comprehensive description here.

L311: please avoid one-sentence paragraphs at all times.
R: will be adapted

L314: what are the assumptions to this two end-member mixing model, and are these assumptions valid in the Rosalia catchment? What is the influence of soil water during rainfall events, and what is the EC signature of soil water vs. groundwater?
Section 5.1: please be more specific and actually quantify the results of your baseflow separation (don't forget to include uncertainties).
L320: please provide a reference for end-member splitting analysis.
L343: please give a measure of how well they match, NSE for instance.
R: L314-343: we will completely re-write section 5. Both examples are probably too complex to be described in this context and we will provide overview presentations in section 5 only. In the meantime, an additional study using the dataset (on the effect of forest access roads on the generation of floods) became available and will be included here. (compare recommendations by Reviewer #2).

L348: please be more specific about the data cleaning process. This is a very important part of the data collection and publication process, and is not mentioned at all in the manuscript.
R: We will remove the statements L233-235 from section 4 and go into more detail here. Please see the reply to the comment on L234.

Table 1: what does 0.2 mm 'events' or 0.1 mm 'events' mean? usually, 0.2 mm is the resolution of individual tips.
R: "0.1 or 0.2 mm events" is the terminology used in the documentation of our tipping bucket rain gauges as well as in the data acquisition system. It refers to rainfall events with a total depth of 0.1 or 0.2 mm. Also, our weighing rain gauges provide an output that simulates a tipping bucket rain gauge with 0.1 mm per tip. We felt that the meaning was clear, but will explain it in the text or in a footnote to the table.

Table 1: Does the "tipping bucket device" have any other specification?
R: Yes. It is listed as 1l (1 liter per tip), but apparently this is difficult to read (In the table, we will write 1 liter/tip). It is used as a complementary device at Q1 to measure discharge when it is smaller than the lower limit of a 1-ft H Flume (0.02 l/s). This never happened since 2015.

Table 1: please also mention the size (i.e., area in ha) of the different sites.
R: We will add size and mean height of the watersheds.

Figure 1: The cities on the inset map of Figure 1 are unreadable, and even the font size of the different sites in the main figure are a bit small. The legend nor caption describes what the green shading or crosshatching indicates. What is a "relais" in this context?
R: we will completely redesign Fig. 1 and remove the relais (the relais just serves for broadcasting data from the RTUs to the base station and is not relevant for using the data)

Figure 3: Please use the metric system.
R: would be misleading. See notes above.

Figure 5: y-axis is too low (Q2 peak cut off).
R: will be corrected

Figure 10: "stream water" or "river water" isotopes rather than river isotopes.
R: will be changed

Figure 11: in its current form, Figure 11 does not add much to the article. The precipitation
and discharge timeseries have already been shown in previous figures, and the
results of the end-member mixing analyses are not shown.
R: in our opinion, this synoptic display of rainfall , discharge and electrical conductivity does add information about the
behavior of the watershed, the Figure might become obsolete in a re-written section 5, as mentioned above.

Figure 12: is this specific discharge or absolute discharge?
R: absolute discharge – as labelled (and will be obsolete or replaced after re-writing section 5).

Morgenschweis, G. (2010) *Hydrometrie - Theorie und Praxis der Durchflussmessung in offenen Gerinnen*,
582 pp., Springer-Verlag, Berlin Heidelberg.

---

## Author Comment (AC2) · 16 Feb 2021

**Josef Fürst (on behalf of the co-authors) Feb 16, 2021**
*Replies are formatted in blue, while original referee's text is black.*

Dear anonymous reviewer,
thank you for your thorough review and your effort to help improve our contribution. I understand your concerns and agree with most of them. I am confident, that we can resolve the issues that you pointed out. Please find detailed replies below.

1. General comments
This data description paper depicts the Rosalia experimental research site in Austria.
It introduces the forested watershed and its characteristics, the monitoring stations and hydrological equipment, the recorded data since 2015, and finally two example studies.
The manuscript describes the sensors and data storage applications in detail, but it should be presented in a more consistent and structured way. In addition, full documentation of the sites and accuracies would be desirable for the understanding of readers and potential users.
R: we agree that the article needs to be improved to be more consistent. We will add more details to the documentation of the sites and on the accuracies of the data.

The two examples give an insight into two aspects of the studies. However, because they are not the main focus of the paper, the explanations and discussions can only be very brief here.
R: We appreciate the comment on section 5. Both examples are probably too complex to be described in this context and we will provide overview presentations in section 5 only. In the meantime, an additional study using the dataset (on the effect of forest access roads on the generation of floods) became available and will be included here.

The datasets are available in the specified data repository. Data collected at the described sites since 2015 are provided. It comprises a documentation of the dataset, GIS and time series data.
2. Specific comments
Right in the third line of the abstract, the operation of the study area since 1875 is mentioned.
The reader looks forward to a long-term data series and analysis. However, he/she is then disillusioned relatively quickly that it is only about the data analysis since 2015. Many graphs even show only two years 2018-2019. I therefore recommend defusing the initially high expectations by moving the long-term aspect from the abstract to the introduction chapter.
R: We agree. We will make clear in the abstract already, that the dataset starts in 2015. The long history of the educational forest of BOKU is still important because many researchers have been working there and have collected data on forestry, vegetation, soils, etc. and gained deep insight into the system.

In order to understand the multiplicity of sites, sensors and measurement data, a comprehensive listing and description is necessary. This is only done partially because the reader has to compile the information himself.
R: we will improve the description of the sites, sensors and data by improved Fig. 1 and Tab. 1 as well as by adaption of the text as described in the detailed replies below.

The following appears to be in need of improvement:
a) Fig. 1 shows sites of 2018, but Tab. 1 shows the status of March 2020. Is the 2018 status up-to-date and does it correspond to the 2020 status?

b) Where is Q2S0 in Fig. 1?
c) The function of R1 Relais (Fig. 1) is not mentioned in the text - is it relevant for
understanding?
R: (a-c): we will completely redesign Fig. 1 to resolve your concerns and improve the text according to the suggestions.
The relais is required for broadcasting between the RTUs, but not required to understand the dataset (to be removed
from Fig. 1).

d) In Tab. 1 there are the sites Q1-4, K1-3, Q2S1 and Q2S2, but Q1S0 and Q2S0 are
missing.
R: the missing sites will be added to Table 1

e) Chapter 3 - L127-137 – is difficult to understand and to match with Tab. 1 and Fig.
1. It would be helpful to insert the site numbers/names here. Otherwise, one has to
pick up everything from these lines and the table and the next chapters.
f) It would also be helpful to add the watershed sizes to Tab. 1. The same applies
also to the depths of the four soil profiles, as these are assigned very unspecifically in
L134-135 and L202-203. A column with the measurement interval and start date of the
sensors used to measure each parameter could also be added to Table 1. To estimate
data quality and sources of uncertainties and errors, further details about the sensors,
such as sensor accuracy and operating range, should be provided with the data. Data
gaps to show the proportion of no-data values could also be visualised in a graph.
R: Thank you for these recommendations! We will improve Fig. 1 and add the requested information to Tab. 1. In an
additional table, we will provide details of the sensors, including sensor accuracy and operating range. Another table or
figure will illustrate the time of records, measurement interval and proportion of no-data values for each site. Also mean
and range of the data values will be included.

g) L127 what is measured: river discharge or water level?
R: The direct sensor output is voltage that is converted to water level. Since both, the H Flume devices and the Thomson
weir at Q3, have a standard geometry with fixed rating curves (not calibrated at site), it is, in my opinion, appropriate to
write about discharge measurement.

h) Chapter 2: It would also be helpful to list the characteristics of the four sub-basins
in more detail: Is there heterogeneity in geology, soils and slopes? Is further information
on soil important for understanding? What are the elevation ranges within the
sub-basins, are there differences between the sub-basins? A map could help for visualisation.
R: In the revised version, we will add the following information: a description of the very uniform geological background,
a soil map (as a figure) and an extended description of the vegetation (per watershed). Elevation ranges will be added to
table 1.

How is the forest managed (maintenance measures, use practices, fertilisation,
sustainability, roads and infrastructures)?
R: We will incorporate the following information into the revised manuscript: The forest management is performed by
the Federal Forests of Austria (Österreichische Bundesforste, OeBf) which is owned by the Republic of Austria. BOKU has
the right of access for educational and research purposes. OeBf claims to manage the forest according to sustainability
principles, balancing protection of environment, the needs of society and commercial success. Management of the forest
is characterized by long production cycles of 100 to 140 years. The main species are the broadleaved beech (fagus
sylvatica) and coniferous Norway spruce (picea abies). Natural regeneration is preferred to planting. Fertilisation almost
never occurs. Timber harvesting is usually done by means of harvesters and forwarders, at steep slopes cable cranes are
used. Management and timber transport are supported by a dense network of forest roads (50m per hectar), suitable for
heavy timber trucks. Main threats are snow break, wind throw and bark beetles, the latter affecting mainly coniferous
tree species.

Chapter 4.1: Is the specific discharge (L245-246) related to site Q3? What about the
other sites?
R: Specific discharge is similar in all watersheds, which will be clarified in the revised version.

As this is a data description paper - add mean and range for all four
gauges. The same for chapter 4.4.

R: We will provide either a separate table or include the information in the table described in the reply to comment f) above.

Chapter 4.5: Which method was used for the isotope analyses in the lab?
R: We used a laser spectroscope (Picarro L2140-i, cavity ring-down spectroscopy). This information will be added.

Chapter 5.2: This is an interesting topic, but too complex for this kind of data description paper. Therefore, some assumptions and relationships are unproven, not supported by numbers or graphs (_L325-336). Exact model performance remains unclear, statistical indicators are missing. Reference to other studies and a discussion are also not provided. Therefore, a separation between an overview presentation in this data description paper and a scientifically sound analysis in an original research article would certainly make more sense.
R: As mentioned earlier, we agree to the comment and give an overview presentation in this paper only.

3. Technical corrections
Fig. 1: German-language city names (Wien, München. . .) appear in Fig. 1; the English names would be appropriate for this map in an English-language paper.
R: Fig. 1 will be redesigned to fulfill this and other requirements.

Fig. 7: Same scale length or axis layout as in Fig. 5 enables a better comparison.
R: we will harmonize the axis style and also correct the y-axis scale in Fig. 5

L314: Add 'electrical' for electrical conductivity.
R: will be added

4. References
Chapter 4.6: What is the source of the DEMs? Add references.
R. references will be added

Reference list: L418-419 Roadmap & Strategy Report on Research Infrastructures – cite as in the text as European Strategy Forum on Research Infrastructures, 2020 or ESFRI, 2020.
R: will be corrected. *This is a bug (or missing document category) in the Endnote style sheet provided by Copernicus.*

Missing references in reference list which can be found in the text: Cosby and Emmett, 2020; Gröning et al. 2012; Hydrologic Engineering Center, 2010; Hipp et al., 2019 Klaus and McDonnell, 2013; Müller et al., 2018; McGuire and McDonnell, 2006; Stevens, 2015.
R: We will add those to the reference list.

5. Data repository
Regarding the file 'Isotope_ESSD.xlsx' in Table 'Q4DailyIso' in the data repository:
Strange or missing values are marked and explained in the column 'Comment'. But gaps of several days are only marked by a line but not by an explanation, e.g. from 25.06.2019 to 03.07.2019, from 16.08.2019 to 28.08.2019, from 20.09.2019 to 04.10.2019, etc.
R: We will clarify all data gaps and include comments. Some gaps resulted from not being able to visit the field site regularly and collecting samples for isotope analysis.

---

## Author Response (AR1)

**Josef Fürst (on behalf of the co-authors) April 26, 2021**
*Replies are formatted in blue, while the original referee's text is in black. Line numbers in blue refer to the revised manuscript.*

Dear anonymous reviewer,
thank you for your thorough review and your efforts to improve our article. We have gratefully adopted the suggestions and made extensive changes to the article. The changes are documented below:

This data publication aims to describe the Rosalia experimental watershed, and introduce the data that is collected between 2015 and 2019. The authors give a very detailed description of all sensors and data storage application used, which I feel came at the expense of more information about the actual watershed and data. The geological background is summarized in one sentence only, and no geologic, vegetation or soil maps are shown, which are key if other researchers are to work with the data.
R: The geological background is indeed very uniform and is described in a few sentences in section 2 (L 99 to 127), which was comprehensively re-written. A soil map with the main soil categories, watershed divides and gauges has been added.

No information is given about the process of data cleaning, or the analysis of the isotopic samples in the lab. Since one of the main aims of a data publication is that other people can work with the data after, I suggest that the article is adapted so that such crucial (!) information is described, and other researchers can also work with the data.
R: Information about data cleaning is introduced in section 4, L 251 – 254, and described specifically in the context of the data-specific paragraphs. Stable isotopes (L 244 to 247): $\delta 18O$ and $\delta 2H$ are analysed using laser spectroscopy (Picarro L 2140-i, Picarro Inc., Santa Clara, CA, USA) in the isotope laboratory at BOKU. A calibration with laboratory reference material calibrated against the Vienna Standard Mean Ocean Water and Standard Light Antarctic Precipitation scale was used. All values are given in delta notation, and the precision of the instrument ($1\sigma$) was better than 0.1‰ and 0.5‰ for $\delta 18O$ and $\delta 2H$.

Some timeseries of the actual data are shown (i.e., of discharge, soil moisture, rainfall, electrical conductivity and the stable isotopic composition of discharge and rainfall), but the presentation of these is very minimal. Furthermore, for or one of the figures the axes were not correctly chosen (i.e., cutting off part of the data), and the figure captions cover only the bare minimum of information.
R: The data repository contains comprehensive documentation and visualization of the time series data in an easily accessible and interactive HTML format. Therefore, we selected only 2 years for the figures in the paper to improve readability. We extended the captions. The characterization of the time series was extended in the text and by new tables including basic statistics. The wrong axis scaling of Fig. 5 (now Fig. 6) was corrected.

When reading this article, I stumbled over numerous grammar mistakes, wrong interpunction, colloquial language, use of the imperial system, and sentences that were clearly not formulated in correct English. I felt like I was doing the final reading before submitting, rather than a review.
R: We removed mistakes and improved the text. It was proof-read by a native speaker in a professional proof-reading service.
Regarding use of the imperial system: this is used only in the naming of the H flume devices where it is, in our opinion, appropriate. H flumes are standardized, off-the-shelf devices which are characterized by their depth in ft. Even European vendors and German textbooks on hydrometry (E.g. (Morgenschweis, 2010)) list H flumes by their depth in ft (and not m). The rating curves for H flumes have been developed by the US SCS for a range of different sizes and therefore it would not help any reader to write about a 0.305 m or 0.61 m H Flume.

I was surprised that this was the case because from
the abstract it sounded like the Rosalia catchment is the flagship of BOKU, and its
documentation thus would deserve adequate attention. In addition to the language of
this article, the structure also clearly needs more time and attention. Some definitions
and topics are introduced but not fully discussed, and come back multiple times in the
manuscript. This does not help the future reader of this article to find the information
needed.
I apologize for my lack of in-depth comments to this article, but this article needs more
time and attention before an in-depth review can be helpful. I suggest that the authors
take this task serious and resubmit after careful re-structuring and rewriting. Documentation
for a long-term research site (1875!) should be more comprehensive than
this, and should for instance also include a background of the most important findings
and the mechanistic understanding of how this watershed functions, in addition to the
missing information with regard to data processing as mentioned above.
R: Mentioning that BOKU is using the Rosalia forest since 1875 was not meant to implicitly indicate that data are available
for the entire time period. To avoid any misunderstanding, we removed or reformulated those parts to clearly focus on
the monitoring network established and the data recorded since 2015.

Detailed comments:
L21: remove additionally
R: removed. New L 20-21: In addition, since 2018, nitrate, TOC and turbidity have been monitored at one gauging station.
In 2019, a programme to collect isotopic data in precipitation and discharge was initiated.

L24: one site of how many sites? The discharge gauging stations?
R: at one new discharge gauging station (Q4). See new L20-21.

L24: nitrate is capitalized where it should not be
R: corrected

L28: remove 'their'
R: corrected

L32: Global change impacts, such as climate warming? I don't see how climate itself
is a global change impact.
R re-phrased L 29-31: Given these long-term datasets, changes in the hydrological cycle, such as those resulting from
climate warming, can be investigated in these watersheds (Bogena et al., 2018).

L33-35: Although I somewhat agree, who realized this? reference needed
R:Re-phrased and reference added (L 32-34): In recent decades, there has been growing recognition that hydrology (and
its related disciplines) cannot be treated in isolation. Rather, hydrological processes driven by meteorological conditions
are also strongly controlled by complex feedback mechanisms with biotic and abiotic systems (Porporato and Rodriguez-
Iturbe, 2002).

L35: experimental catchments? remove sites
R: corrected: L34-35: Therefore, hydrological experimental watersheds have gradually transitioned into multi-disciplinary
experimental watersheds.

L45: unclear which framework is referred to
R: re-phrased (L 40-45): Examples of such networks are the German 'TERrestrial ENvironmental Observatory network'
(TERENO) (Zacharias et al., 2011), the 'International Network for Alpine Research Catchment Hydrology' (Bernhardt et al.,
2015), the 'US National Science Foundation's National Ecological Observatory Network' (NEON) (Kampe et al., 2010), and
the 'Euro-Mediterranean Network of Experimental and Representative Basins'(ERB) as part of UNESCO FRIEND (Flow
Regimes from International Experimental and Network Data) (Holzmann, 2018).

L57-65: why is the LTER not introduced together with the other networks?
R: LTER is now introduced together with the other networks (L46-54)

L74: if the object "was and still is" the word "is" is sufficient to indicate that
R: re-phrased (L75): "The overall objective is to implement …"

L76: if this is "a research emphasis" what are other important points?
L79: what are 'point related measurements'?point measurements?
L76-80: please rephrase this sentence to provide more clarity.
R: re-phrased L75-78: The overall objective is to implement a multi-scale, multi-disciplinary observatory that facilitates the study of water, energy, and solute transport processes in the soil-plant-atmosphere continuum. Research emphasis is put on deriving effective parameters for scales on which models simulate flow and transport processes (e.g. hillslope, catchment) by upscaling point measurements.

L83: how does the set-up allow for these experiments, in comparison to other sites?
R: BOKU has management options in the forest that are generally not available in private forests where often just the implementation of monitoring stations is possible.
We added the following information into the revised manuscript (L80-86): Because BOKU has the right of access for educational and research purposes, large-scale controlled experiments can be undertaken. For example, rain-out shelters were used in parts of the forest by Netherer et al. (2015) to investigate drought impacts on bark beetle attacks on Norway spruce, while Schwen et al. (2015) and Leitner et al. (2017) used rain-out shelters to investigate soil water repellency and short-term organic N-fluxes under a changing climate. Besides such local experiments, the monitoring network established since 2015 enables researchers to investigate the impacts on the large-scale forest ecosystem and its services by providing the necessary baseline data. Investigating the transition of the forest ecosystem from its actual state into a pristine, unmanaged natural forest is among future research plans.

L89: "are and will be investigated by a team of researchers" this sounds as if the team is already chosen, and cannot be adapted anymore. This is contrary to what I would expect is the aim of publishing this article, which is to promote other researchers to also use the data that is being published in this publication.
R: With publishing the dataset we are certainly inviting other researchers to use the data too. We rephrased L 80-86 (see above).

L89: same comment as with "was and still is" in L77
R: re-phrased, see above.

L95-98: since this is such a standard article lay-out, I would suggest that the others consider removing this description.
R: removed

L100-101: this sentence is gramatically incorrect.
L102: 'is' steeper than
R: corrected (L 91-93): Terrain heights range from 320 to 725 m a.s.l., and the watershed is characterised by steep slopes (96% of the area is steeper than 10%, and 55% steeper than 30%).

L111-112: gramatically incorrect sentence
R: comprehensively re-written paragraph.

**The description of the watershed was extended to provide more information on geology, soils, vegetation and forest management(L 99-127), and a soil map was added (Fig. 3)**

L129: the names of the watersheds, and their respective sizes, have not been introduced yet.
R: we improved this part and redesigned Fig. 1 and Table 1 to provide this information.

L30: monitored "with" a spectrometer probe
R: corrected (L 142)

Line 131-135: every sentence starts on a new line.
R: corrected

L312(132?): which altitudes?
R: altitudes were added to table 1. Also, Fig. 1 displays terrain height.

L136-137: this sentence is redundant because this is mentioned in the figure and table captions.
R: removed

L148: please specify what the "DMBS addVANTAGE Pro' is directly when first mentioning it.
R: short description of addVANTAGE Pro was added (L 160 – 166).

L154: can the authors be more specific about the treatment samples after being collected
by the totalisers or as grab samples? How are these samples stored in the
samplers to ensure that the chemistry and isotope samples can both be analyzed adequately?
R: we described the sampling in L 167-169: Stable water isotope data are not automatically uploaded to the DBMS but
samples are collected on-site and picked up manually by university staff for analysis in the laboratory. Precipitation
samples are collected bi-weekly with totalisators with plans to refine the sampling interval to daily, while streamflow
samples are collected as daily grab samples using an autosampler.

Handling of the samples and laboratory analysis is comprehensively described in section 3, L222-247.

157: The field courses are organized by students? Or should this be "by students
during field courses
R: moved to section 5 Applications (L 355-358): The site is regularly used for advanced field courses in the water
management and environmental engineering curriculum. During these courses, students not only learn about the setup
and operation of a hydrological monitoring network, but they also contribute to the improvement of knowledge about
the watershed by collecting and analysing soil samples or performing validation measurements of the instruments.

L158: which other (LiDAR-based) DEMS are available? and, LiDAR is commonly
spelled with a lower-case 'i'
R:  reference to DEMs moved to section 4.6 Spatial data (L 348-352). We kept the spelling LIDAR, which is as common as
LiDAR.

L161: what is a "hydrological" site? A site at which hydrological measurements are
being performed? in this case, the word 'hydrological' is redundant, given the sentence
that follows.
R: re-phrased (L172): The sites for discharge measurements were selected to collect data for nested sub-catchments of
different sizes.

L163: new line started where not needed.
R: corrected

L169: grammatically incorrect sentence. L170: grammatically incorrect sentence
R: checked and corrected

L168: please use the metric system.
R: Foot is appropriate in this context, see note above

L178: Reference missing for the "Thompson" weir.
R: reference added (L185)

L181: is their SDI-12 interface really important to mention in this article? And if so, be
specific as to why the SDI-12 interface is preferred.
R: it is not directly relevant for the dataset, but readers might be interested in the sensors that we use. SDI-12 sensors are
generally known for their very low power requirements and their standard interface to most data-loggers.

L181-184: Colloquial language. Please rewrite.
L183-184: please rewrite to make the sentence clearer.
R: re-phrased L191-196: They work electronically reliably, but the measured conductivities are sensitive to biofilms on the
sensor, and the internal firmware requires more than an hour to achieve a stable reading after power-on or after
cleaning. Furthermore, the measured conductivity tends to show an offset compared to manual measurements
conducted approximately bi-weekly. Nevertheless, the recorded curves show plausible dynamics, e.g., during storm
events. Currently, alternative sensors are tested to replace the C4E devices. At site Q4, a different type of sensor (s::can
condu::lyser ™) is used, which, after more than a year of operation, recorded reliable and stable data.

L189: should be "are' possible.
R: corrected (L199)

L190: atmospheric deposition of what? Salts, leaves? please be specific.

R: re-phrased L199-201: They require more maintenance than weighing rain gauges because the funnel is easily blocked by deposition of leaves, pollen, dust or insects, and they are inoperable during frost.

L193: please rewrite to clarify the meaning of the sentence. Also, please quantify and be specific about how the rain measurements are affected, and why they are reliable in this data publication.

R: re-phrased L 202- 205: Also, in the forest, it was not possible to follow all the rules for the proper placement of a rain gauge. Particularly, the recommendation that the height of nearby objects, such as trees, should not exceed the distance from the gauge to the objects (WMO, 2008), had to be disregarded for Q1 and Q2. In particular, the rain gauge at Q1 is directly affected by the interception of the trees above.

Added discussion in section 4.2 (L 304-311): In this densely forested watershed, it is not possible to place all rain gauges at sites without interception or rain-shading. However, a comparison of rainfall depths at all seven rain gauges for several events revealed good agreement. Gauge Q1 is affected by interception, which amounts to typically less than 2 mm per event (compared to weighing rain gauges K1 and K2), but monthly precipitation at Q1 is on average only 75% of the mean of K1 and K2. At Q2, monthly precipitation is on average 87% of the mean of K1 and K2. (K1 is close to the highest elevation of the watershed, K2 at the lowest – see Figure 1 and Table 1). Therefore, the data from all rain gauges are useful for analysing storm events, as interception reduces rainfall depths by only a small percentage. For water balance investigations of periods longer than a week, however, only the gauges not affected by interception should be used

L212: $d18O$ and $d2H$ are already defined earlier in the manuscript. Please use the short-hand notation to make the text more concise, or refrain from defining the shorthand notations.

R: corrected (L223)

L230: 'using' addVantage Pro?' or does the program also assess the data? If so, please be specific about which protocols are used.

R: changed to "using" (L249).

L234: can the authors be more specific about this data cleaning process?

R: variable-specific statements on data cleaning have been added

**Discharge data (L 265-272)**: Raw discharge data at the H flume gauges Q1, Q2, and Q4 needed careful inspection and editing. First, spikes in the hydrographs (one or two consecutive values significantly exceeding the value before and after the spike) were attributed to random events such as a leave under the ultrasonic depth sensor and were automatically replaced by linear interpolation. Next, visually detected implausible discharges were replaced by linear interpolation where reliably possible, or deleted otherwise. As an example, occasionally during very low flow, single leaves can temporarily (a few hours) get stuck at the narrow outlet of the flume and cause the water level to rise a few millimetres. Such events are clearly visible as plateau-shaped parts of the hydrograph and can be safely replaced by linear interpolation. At these gauges, the measurements have never disturbed by freezing.

At the weir Q3, two issues required editing: 1) during very low flow, leaves and grass can occasionally get stuck at the weir crest, causing the water level to rise. These events can be detected in the images transmitted daily by a surveillance camera and visually in the hydrograph. Such artefacts are replaced by linear interpolation; 2) during longer frost periods, the stilling basin may be covered by ice and therefore the discharge is no longer described by the weir formula. These situations can be detected by visual inspection of the hydrograph and comparison with the temperature. These parts of the records have been deleted.

**Precipitation (L293-299)**: For quality control, rainfall data recorded by tipping bucket devices (Q1 to Q4) are compared to records of the weighing rain gauges and to corresponding hydrographs. They are deleted if the funnel appears to have been (partially) blocked. Also, records for the winter season from November to February are excluded due to tipping bucket issues with freezing. Anomalies observed during field maintenance visits (one to two per month) are also considered. The three weighing rain gauges have provided gap-free records since the time of installation up to now, with a resolution of 0.1 mm. For most rainfall events between March and October, consistent and plausible data were acquired by up to seven rain gauges in total, providing a high-resolution rainfall pattern for a small area of 222 ha, and being spread over different altitudes from 320 to 700 m a.s.l (Table 4, Figure 8).

**Electrical conductivity (L 328-333)**: Besides frost, conductivity records at sites Q1 and Q2 are additionally negatively influenced by the sensor problems described in section 3.2. Regular conductivity measurements with a portable device showed that the conductivity of base flow is stable at sites Q1 and Q2 (typically approx. 120 µS/cm), so that the recorded conductivity series are still informative for the separation of base flow and direct runoff events, despite conductivity offsets in the records.

L247: redundant to describe what Figure 5 illustrates, because this is mentioned in the caption. Please refer to the figure in the text itself.
R: corrected

L247: hydrographs 'for' July and August 2018
R: corrected (L286).

L273: could it not also be due to natural preferential flow paths? and if not, why not? and since in L274 the natural pref. flowpaths are mentioned, please be more specific about the limits to the period at which the disturbance affected the measurements.
R:extended the explanation (L316-322): It is important to mention that the installation of the sensors requires digging a trench, which causes considerable local disturbance of the soil. Despite careful refilling, local infiltration paths could be influenced, and data do not necessarily reflect natural conditions for some time after installation. During the first few months after installation, for example, deeper probes reacted faster to rainfall than those close to the surface (Figure 10). This can be attributed to artificial flow paths along the walls of the trench and the cables, or to effects arising from interrupted and destroyed natural macropores like wormholes. However, direct effects due to installation practically disappeared after the first season.

L288: I would expect to find this sentence in an introduction, not in a 'results' section
R: removed

L298: reference?
R: added reference (L 340): Feng et al. (2009 ): "Seasonality of isotopes in precipitation: A global perspective"

L300: which stable isotope? oxygen18 I assume?
R: $\delta^{18}O$ added in L 341

L305: I think the spatial data can be introduced where the DEMS are introduced first, and don't need a separate section dedicated to them.
R: It is more than just DEM, also watershed divides, surveyed creeks and location of sites. We add a more detailed description here (L 348-352): Data interpretation is complemented by a comprehensive amount of spatial data characterising the site. DEMs at various resolutions are available, including a 10 × 10 m DEM provided by the government of Austria, and a LIDAR based DEM at 0.5 × 0.5 m (Immitzer, 2009), accessible at https://zenodo.org/record/4601057. From these DEMs, watershed divides and the drainage network were derived in GIS. Additionally, a ground survey was performed for the main creeks in 2018. These data are included in the repository in shapefile format.

L311: please avoid one-sentence paragraphs at all times.
R: Section 5 is entirely re-written (L 354-374)

L314: what are the assumptions to this two end-member mixing model, and are these assumptions valid in the Rosalia catchment? What is the influence of soil water during rainfall events, and what is the EC signature of soil water vs. groundwater?
Section 5.1: please be more specific and actually quantify the results of your baseflow separation (don't forget to include uncertainties).
L320: please provide a reference for end-member splitting analysis.
L343: please give a measure of how well they match, NSE for instance.
R: we completely re-wrote section 5. Both examples are probably too complex to be described in this context and we provide overview presentations in section 5 only. In the meantime, an additional study using the dataset (on the effect of forest access roads on the generation of floods) became available and is included here. (compare recommendations by Reviewer #2).

L348: please be more specific about the data cleaning process. This is a very important part of the data collection and publication process, and is not mentioned at all in the manuscript.
R: we paid attention to the data cleaning process, which is described in section 4 and its sub-sections.

Table 1: what does 0.2 mm 'events' or 0.1 mm 'events' mean? usually, 0.2 mm is the resolution of individual tips.
R: "0.1 or 0.2 mm events" is the terminology used in the documentation of our tipping bucket rain gauges as well as in the data acquisition system. It refers to rainfall events with a total depth of 0.1 or 0.2 mm. Our tipping bucket devices

have a resolution of 0.2 mm, our weighing rain gauges provide an output that simulates a tipping bucket rain gauge with 0.1 mm per tip. We record a time series of the tipping times so that the highest possible temporal resolution of precipitation intensity is obtained. These data are not included in the data repository and the reference has been removed from Table 1.

Table 1: Does the "tipping bucket device" have any other specification?
R: Yes. It is listed as 1l (1 liter per tip), but apparently this is difficult to read. It is used as a complementary device at Q1 to measure discharge when it is smaller than the lower limit of a 1-ft H Flume (0.02 l/s). This never happened since 2015. In table 1 it reads "1 liter per tip" now.

Table 1: please also mention the size (i.e., area in ha) of the different sites.
R: We added the sizes of the watersheds and height of sites.

Figure 1: The cities on the inset map of Figure 1 are unreadable, and even the font size of the different sites in the main figure are a bit small. The legend nor caption describes what the green shading or crosshatching indicates. What is a "relais" in this context?
R: we completely redesigned Fig. 1 and removed the relais (the relais just serves for broadcasting data from the RTUs to the base station and is not relevant for using the data)

Figure 3: Please use the metric system.
R: would be misleading. See notes above.

Figure 5: y-axis is too low (Q2 peak cut off).
R: corrected (now Fig 6). Fig. 6 displays Q in log scale now to improve readability

Figure 10: "stream water" or "river water" isotopes rather than river isotopes.
R: changed in new Fig. 11

Figure 11: in its current form, Figure 11 does not add much to the article. The precipitation and discharge timeseries have already been shown in previous figures, and the results of the end-member mixing analyses are not shown.
R: the Figure became obsolete in the re-written section 5, as mentioned above.

Figure 12: is this specific discharge or absolute discharge?
R: the Figure became obsolete in the re-written section 5, as mentioned above.

Morgenschweis, G. (2010) *Hydrometrie - Theorie und Praxis der Durchflussmessung in offenen Gerinnen*, 582 pp., Springer-Verlag, Berlin Heidelberg.

*Replies are formatted in blue, while the original referee's text is in black. Line numbers in blue refer to the revised manuscript.*

Dear anonymous reviewer,
thank you for your thorough review and your efforts to improve our article. We have gratefully adopted the suggestions and made extensive changes to the article. The changes are documented below:

1. General comments
This data description paper depicts the Rosalia experimental research site in Austria.
It introduces the forested watershed and its characteristics, the monitoring stations and
hydrological equipment, the recorded data since 2015, and finally two example studies.
The manuscript describes the sensors and data storage applications in detail, but it
should be presented in a more consistent and structured way. In addition, full documentation
of the sites and accuracies would be desirable for the understanding of
readers and potential users.
R: we improved the text to be more consistent. We added more details to the documentation of the sites and on the accuracies of the data (new Table 2).

The two examples give an insight into two aspects of the studies. However, because
they are not the main focus of the paper, the explanations and discussions can only be
very brief here.
R: We have completely revised chapter 5. It now contains a list and very brief description of the previous applications.

The datasets are available in the specified data repository. Data collected at the described
sites since 2015 are provided. It comprises a documentation of the dataset,
GIS and time series data.
2. Specific comments
Right in the third line of the abstract, the operation of the study area since 1875 is mentioned.
The reader looks forward to a long-term data series and analysis. However,
he/she is then disillusioned relatively quickly that it is only about the data analysis since
2015. Many graphs even show only two years 2018-2019. I therefore recommend defusing
the initially high expectations by moving the long-term aspect from the abstract
to the introduction chapter.
R: We re-wrote the abstract to avoid any confusion about the time period of the dataset (L12-23).

In order to understand the multiplicity
of sites, sensors and measurement data, a comprehensive
listing and description is necessary. This is only done partially because the
reader has to compile the information himself.
R: we improved the description of the sites, sensors and data by improved Fig. 1 and Tab. 1. Table 2 was added to provide characteristics of the sensors. We modified the text as described in the detailed replies below.

The following appears to be in need of improvement:
a) Fig. 1 shows sites of 2018, but Tab. 1 shows the status of March 2020. Is the 2018
status up-to-date and does it correspond to the 2020 status?
b) Where is Q2S0 in Fig. 1?
c) The function of R1 Relais (Fig. 1) is not mentioned in the text - is it relevant for
understanding?

R: (a-c): we completely redesigned Fig. 1 to resolve your concerns and improved the text according to the suggestions. The relais is required for broadcasting between the RTUs, but not required to understand the dataset (removed from Fig. 1).

d) In Tab. 1 there are the sites Q1-4, K1-3, Q2S1 and Q2S2, but Q1S0 and Q2S0 are missing.
R: the missing sites were added to Table 1 and clearly labeled in Fig 1

e) Chapter 3 - L127-137 – is difficult to understand and to match with Tab. 1 and Fig. 1. It would be helpful to insert the site numbers/names here. Otherwise, one has to pick up everything from these lines and the table and the next chapters.
f) It would also be helpful to add the watershed sizes to Tab. 1. The same applies also to the depths of the four soil profiles, as these are assigned very unspecifically in L134-135 and L202-203. A column with the measurement interval and start date of the sensors used to measure each parameter could also be added to Table 1. To estimate data quality and sources of uncertainties and errors, further details about the sensors, such as sensor accuracy and operating range, should be provided with the data. Data gaps to show the proportion of no-data values could also be visualised in a graph.
R: We improved Fig. 1 and added the requested information to Tab. 1. In an additional table 2, we provide details of the sensors, including sensor accuracy and operating range. Tables are added in the sections of the data description to illustrate the time of records, and proportion of no-data values for each site. Also mean and range of the data values are included.

g) L127 what is measured: river discharge or water level?
R: The direct sensor output is voltage that is converted to water level. Since both, the H Flume devices and the Thomson weir at Q3, have a standard geometry with fixed rating curves (not specifically calibrated at site), it is, in our opinion, appropriate to write about discharge measurement.

h) Chapter 2: It would also be helpful to list the characteristics of the four sub-basins in more detail: Is there heterogeneity in geology, soils and slopes? Is further information on soil important for understanding? What are the elevation ranges within the sub-basins, are there differences between the sub-basins? A map could help for visualisation.
R: we added the following information: a description of the very uniform geological background, a soil map (Fig. 3) and an extended description of the vegetation (per watershed). Fig. 1 displays the terrain heights. Elevation of the gauges was added to table 1.

See section 2, L89-127

How is the forest managed (maintenance measures, use practices, fertilisation, sustainability, roads and infrastructures)?
R: We added the following information into the revised manuscript (L 119-127): Forest management is performed by the Austrian Federal Forests (Österreichische Bundesforste, OeBf) owned by the Republic of Austria. BOKU has the right of access for educational and research purposes. OeBf manages the forest sustainably, balancing the protection of the environment, the needs of society, and economic success. The management of the forest is characterised by long production cycles of 100 to 140 years. The main species of the forest are the broadleaved beech (fagus sylvatica) and the coniferous Norway spruce (picea abies). The forest is at different development stages ranging from clear cut areas to mature forest stands. Natural regeneration is preferred to planting, and fertilisation is almost never done. Timber harvesting is usually done with harvesters and forwarders, and cable cranes are used at steep slopes. Management and timber transport are supported by a dense network of forest roads (50 m per hectare), suitable for heavy timber trucks. Main threats to the forest are snow break, wind throw and bark beetles, the latter affecting mainly coniferous tree species.

Chapter 4.1: Is the specific discharge (L245-246) related to site Q3? What about the other sites?
R: rephrased L280-282: Specific discharge does not vary significantly between the four watersheds and typically ranges from 1 to 2 l s$^{-1}$ km$^{-2}$ during low to medium flow and up to 30 l s$^{-1}$ km$^{-2}$ during peak flows (calculated from daily means).

As this is a data description paper - add mean and range for all four gauges. The same for chapter 4.4.

R: We added Tables 3 and 4 to provide the requested information.

Chapter 4.5: Which method was used for the isotope analyses in the lab?
R: We used a laser spectroscope (Picarro L2140-i, cavity ring-down spectroscopy). This information was added in section 3.2, sub-section water quality (L244-247). The description of collecting and pre-treating the samples was also extended (L 222-243).

Chapter 5.2: This is an interesting topic, but too complex for this kind of data description paper. Therefore, some assumptions and relationships are unproven, not supported by numbers or graphs (_L325-336). Exact model performance remains unclear, statistical indicators are missing. Reference to other studies and a discussion are also not provided. Therefore, a separation between an overview presentation in this data description paper and a scientifically sound analysis in an original research article would certainly make more sense.
R: Section 5 is entirely re-written (L 354-374).

3. Technical corrections
Fig. 1: German-language city names (Wien, München. . .) appear in Fig. 1; the English names would be appropriate for this map in an English-language paper.
R: Fig. 1 was redesigned to fulfill this and other requirements.

Fig. 7: Same scale length or axis layout as in Fig. 5 enables a better comparison.
R: we harmonized the axis style and also corrected the y-axis scale in Fig. 5 (now Fig 6).

L314: Add 'electrical' for electrical conductivity.
R: added in L228

4. References
Chapter 4.6: What is the source of the DEMs? Add references.
R. references added (L348-352): DEMs at various resolutions are available, including a 10 × 10 m DEM provided by the government of Austria, and a LIDAR based DEM at 0.5 × 0.5 m (Immitzer, 2009), accessible at https://zenodo.org/record/4601057. (Publication of this DEM on Zenodo was triggered by this dataset publication!)

Reference list: L418-419 Roadmap & Strategy Report on Research Infrastructures – cite as in the text as European Strategy Forum on Research Infrastructures, 2020 or ESFRI, 2020.
R: all references have been carefully checked and corrected.

Missing references in reference list which can be found in the text: Cosby and Emmett, 2020; Gröning et al. 2012; Hydrologic Engineering Center, 2010; Hipp et al., 2019 Klaus and McDonnell, 2013; Müller et al., 2018; McGuire and McDonnell, 2006; Stevens, 2015.
R: added. Klaus and McDonnell, 2013; and McGuire and McDonnell, 2006 became obsolete.

5. Data repository
Regarding the file 'Isotope_ESSD.xlsx' in Table 'Q4DailyIso' in the data repository:
Strange or missing values are marked and explained in the column 'Comment'. But gaps of several days are only marked by a line but not by an explanation, e.g. from 25.06.2019 to 03.07.2019, from 16.08.2019 to 28.08.2019, from 20.09.2019 to 04.10.2019, etc.
R: data gaps were clarified and comments included. Some gaps resulted from not being able to visit the field site regularly and collecting samples for isotope analysis. An updated file was uploaded to the repository.

---

## Referee Report (RR1)

This data publication describes the Rosalia experimental watershed, and introduces the data that is collected between 2015 and 2019. The manuscript is a revised version, and has greatly improved from its first submission. The authors added background information about the watershed such as soil and vegetation information, and were much more explicit about data cleaning processes, which will be crucial for future users of the data. I appreciate the additional figures and improvements on the figures, which are much more informative and readable now.

Although the article has clearly improved I still found that sometimes the text was longer than it needed to be. One reason for that could be that some topics (e.g., rain gauges and rainfall sampling) are discussed in different paragraphs. Hence, the reader had to be reminded of the rain gauge set-up to make the paragraph understandable. Re-structuring the article a bit more such that all relevant information is mentioned together will further reduce the length of the article and improve its readability. It will also reduce questions such as "how is precipitation sampling protected against fractionation" that arose as I first read about the rain gauges.

I found the summary rather confusing. It starts with "The data represent an effort..", but which data, "The data presented in this manuscript .." ? being more specific would be helpful here. "The record for some components started in 2015" also doesn't say much. When did it start for other components? And for which components did it start in 2015? Again, making sentences more specific will increase the readability and avoid frustration with the reader (e.g., "The first monitoring stations for precipitation, discharge and weather variables were installed in 2015, and xx additional nested catchments have been added since then."). Then, the summary goes on to explain why it's important to publish data and then it moves back to a sentence about the measurements (data cleaning), showing that also here the information is scattered rather than presented in one location. I suggest rewriting the summary such that topics are described together and such that it reflects the content of the paper is needed.

Minor comments:

L35: CZO, update with more recent CZN
L56: giving examples of these questions would be more meaningful
Fig.1: Why are climate stations (K1 and K2) and soil moisture stations located at the same elevation, rather than covering a range in elevation?
L77: examples of these parameters?
L105: "water holding capacity" rather than wind capacity? And please be specific about how wind and slope affect organic content. "wind" doesn't do much to organic content, but wind erosion might. Slope doesn't do much to the organic content, either, but gravitational transport does.
L09 onwards: maybe a more descriptive name than category 2 is possible, and would make it easier for the reader to follow.
Fig. 3: use other numbers to indicate category's – currently confusing with percentages.
L142: spectrometer probe, mention brand or refer to table
Table 2: not sure if the websites here are needed…
L169: (how) are the autosamplers equipped to avoid fractionation?
L203: maybe include the 'rules for proper placement' or at least the rules that were violated (in addition to the one that is mentioned particularly.
L205: how are these rain gauges equipped to avoid fractionation?
L217: refer to figure 1?
L227: how are these samples protected from fractionation when left without maintenance for 24 days, particularly when freezing issues occur (L227-230).
L231: please explain more carefully. The grab sample was analyzed directly or kept in a closed bottle until the autosampler sample was analyzed (open for 24 days?). Please show these results here or in the supplementary material, or at least mention an $r^2$ value or any other type of statistics.

Sections rain gauges and water quality: please merge these sections such that all relevant information is presented together. In the current version, the sites and type of rain gauge are repeated, and reading the initial rain gauge section raises questions about sampling is done.

L267: was a time-window (smoothing window) applied for the linear interpolation? If so, please mention the time-window.

L269 and L273: please define 'very low flow'

L285: increased compared to what?

L304: please merge with other paragraph describing this problem.

L305: good agreement of what? The timing? Or the precipitation magnitude? Why would precipitation magnitude be similar at two different stations?

---

## Author Response (AR2)

**Anonymous Referee #1, Report #2**
Submitted: 26 May 2021

**Josef Fürst (on behalf of the co-authors) June 28, 2021**
*Replies are formatted in blue, while the original referee's text is in black. Line numbers in blue refer to the revised manuscript.*

Dear anonymous reviewer,
thank you for your second thorough review and your additional efforts to improve our article. We have gratefully adopted the suggestions and made changes to the article. The changes are documented below:

Although the article has clearly improved I still found that sometimes the text was longer than it needed to be. One reason for that could be that some topics (e.g., rain gauges and rainfall sampling) are discussed in different paragraphs. Hence, the reader had to be reminded of the rain gauge set-up to make the paragraph understandable. Re-structuring the article a bit more such that all relevant information is mentioned together will further reduce the length of the article and improve its readability. It will also reduce questions such as "how is precipitation sampling protected against fractionation" that arose as I first read about the rain gauges.

We want to clearly distinguish between precipitation measurement and water quality monitoring. Therefore, the sentence on the Palmex totalisators was moved to section "water quality".

We added the information to the totalisators that they were specifically designed to protect against isotope fractionation.

Minor comments:

L35: CZO, update with more recent CZN

R: added "… and has been succeeded as the Critical Zone Collaboration Network (CZN) since 2021."

L56: giving examples of these questions would be more meaningful

R: there are 23 unsolved problems listed in the reference. In our opinion, the relevant questions for this article are sufficiently summarized in the following sentences on L56-59.

Fig.1: Why are climate stations (K1 and K2) and soil moisture stations located at the same elevation, rather than covering a range in elevation?

R: This comment is not clear: K1 is at 640 m a.s.l. and K2 at 385 m a.s.l. Soil moisture station Q1S0 is next to Q1 at 560 m a.s.l. Soil moisture stations Q2S0 to Q2S2 are next to Q2 and form a slope transect with heights from 558 to 572 m a.s.l. as can be read in table 1 and Fig 1. *(no changes in the manuscript)*

L77: examples of these parameters?

R: added "(e.g., hydraulic conductivity and porosity of soils, soil water movement),"

L105: "water holding capacity" rather than wind capacity? And please be specific about how wind and slope affect organic content. "wind" doesn't do much to organic content, but wind erosion might. Slope doesn't do much to the organic content, either, but gravitational transport does.

R: improved the sentence: "These sites are characterised by poor water holding capacity and loss of organic material due to gravitational transport and wind erosion."

L109 onwards: maybe a more descriptive name than category 2 is possible, and would make it easier for the reader to follow.

R: The names of the soil categories are given at the beginning of the sentences ("Cambisols at plains and moderate slopes"). The single digit category numbers are kept to maintain a consistent link between labels in the map and the legend in Fig. 3.

Fig. 3: use other numbers to indicate category's – currently confusing with percentages.

R: changed the style of the piechart labels to "1: 5%", etc.

L142: spectrometer probe, mention brand or refer to table

R: added S::can multi::lyser$^{TM}$

Table 2: not sure if the websites here are needed…

R: a previous reviewer wanted more details on sensors

L169: (how) are the autosamplers equipped to avoid fractionation?

R: The focus of this paragraph lies on the data collection system, which operates automatically and allows remote access to the data. We only mention the isotope measurements here, as contrary to the rest of the data, their data is not automatically imported into a database, nor is it remotely accessible. Describing the stable isotope measurement systems and their respective protection measures against evaporation in detail would thus not fit into the context. This information is presented later in the manuscript when describing the specific measurements.

L203: maybe include the 'rules for proper placement' or at least the rules that were violated (in addition to the one that is mentioned particularly.

R: The WMO guide to hydrological practice (WMO, 2008) describes appropriate rain gauge location on more than 2 pages. To keep the text short – as recommended – we rephrased to "Furthermore, it was not possible to place all rain gauges in the forest in such a way that no negative wind influences occur. Particularly, the recommendation that the height of nearby objects, such as trees, should not exceed the distance from the gauge to the objects (WMO, 2008), had to be disregarded for Q1 and Q2."

L205: how are these rain gauges equipped to avoid fractionation?

R: To avoid confusion between rainfall measurement and sampling for isotope analysis, the reference to rainfall collectors has been moved to the water quality section. There, we added the information that the samplers are specifically designed to prevent isotope fractionation.

L217: refer to figure 1?

R: added reference to Fig. 1

L227: how are these samples protected from fractionation when left without maintenance for 24 days, particularly when freezing issues occur (L227-230).

L231: please explain more carefully. The grab sample was analyzed directly or kept in a closed bottle until the autosampler sample was analyzed (open for 24 days?). Please show these results here or in the supplementary material, or at least mention an r2 value or any other type of statistics.

R: We now describe the system in more detail and show that the average offset between grab sample and autosampler isotope values was close to the measurement uncertainty of stable isotope ratios.

Sections rain gauges and water quality: please merge these sections such that all relevant information is presented together. In the current version, the sites and type of rain gauge are repeated, and reading the initial rain gauge section raises questions about sampling is done.

R: We did not merge the sections but removed "rainfall sampling" from "rain gauges" to the water quality section to clearly distinguish these two different types of observation.

L267: was a time-window (smoothing window) applied for the linear interpolation? If so, please mention the time-window.

R: as all time series are recorded at a 10-min interval (see section 3.1), the spikes described here consist of one or two values only and therefore the time window for linear interpolation is either 20 or 30 minutes.

L269 and L273: please define 'very low flow'

R: added "(water level less than 2 cm in the flume),"

L285: increased compared to what?

R: rephrased to "In the hydrographs for the period 2018 to 2019 (**Fehler! Verweisquelle konnte nicht gefunden werden.**) it can be seen that the base flow is greater in spring and early summer than in autumn and winter, and that sharp runoff peaks occur after rainfall events."

L304: please merge with other paragraph describing this problem.

R: we feel that this short reference to the problem of rain gauge location is necessary here to introduce the discussion on the following lines.

L305: good agreement of what? The timing? Or the precipitation magnitude? Why would precipitation magnitude be similar at two different stations?

R: we rephrased to "However, the rainfall depths at all seven rain gauges are very similar for larger rainfall events that extend over the whole watershed." In this small watershed of 222 ha only, only minor differences between rainfall depths for time intervals greater than a few hours are to be expected. The short time interval of 10 min allows the spatio-temporal development of a precipitation event to be recorded.
*Replies are formatted in blue, while the original referee's text is in black. Line numbers in blue refer to the revised manuscript.*

Dear anonymous reviewer,
thank you for your thorough review and your efforts to further improve our article. We have gratefully adopted the suggestions and made changes to the article. The changes are documented below:

Thank you for revising the manuscript. From my point of view, it has improved significantly. The description of the watershed, sites and sensors are more detailed and therefore better understandable. Also the added details about data description, accuracy and data cleaning process support the understanding for the readers and possible data users.

However, I still have a few minor comments.
With regard to the Revised Manuscript:
- Fig 1: Source for the DEM / elevation map is missing in the figure caption.

R: added source data.noe.gv.at

- Line 91: This coordinate LAT 47°42'N, LON 16°17'E refers to a single point. However, the corresponding sentence belongs to the watershed description. Either further coordinates would have to be added here to delineate the watershed or it would have to be added which point this is.

R: replaced by reference to Fig 1, which has an overview map showing Lat/Lon graticules.

- Line 92: "Terrain heights range from 320 to 725 m a.s.l.". Why does the map legend in Fig. 1 starts with 361 m asl? Is this due to the map resolution?

R: corrected. Thank you for this careful check! 320 is the lowest elevation of the whole educational forest, while 385 m is the lowest elevation of the watershed.

- Lines 93-94: On what source is this precipitation and temperature data based?

R: added the open data source available via LTER and adjusted the time span.

- Fig. 3: Source for the map is missing in the figure caption.

R: the map has been created for this manuscript, no citeable source.

- Fig. 7: It is pointed out in the figure caption that the highest peaks have been cut off. Nevertheless, it might be interesting for the reader to know when the highest peaks appeared. Would it perhaps make sense to use arrows to mark the positions? Or you can refer to the repository.

R: the full range of the discharges is visible in Fig. 6. The detail in Fig. 7 focuses on the diurnal fluctuations.

- Line 349: Please add a reference of the DEM to the reference list.

R: added

- Line 467: Add that it is a diploma thesis comparable to line 508.
- Line 522: Add that it is a PhD thesis comparable to line 508.

R: consistently added

With regard to the Repository:
In the modified manuscript there are confusions regarding the repository.
In the modified version now there is a new link to https://doi.org/10.5281/zenodo.3997140.
Checking the README.txt file, it states that "This is the repository 'Rosalia: an experimental research site to study hydrological processes in a forest catchment - data repository', available at https://doi.org/10.5281/zenodo.3997141 (Fürst et al., 2020)". This is not consistent because it refers to the old link of the first submitted version.

R: the DOI concept of Zenodo may be a bit confusing, because the last digit is used for versioning. The link https://doi.org/10.5281/zenodo.3997140 is the stable one, because it automatically refers to the newest version. This is preferred, because we intend to update the repository annually. We updated the README.txt file accordingly.

It also includes the still unchanged title of the manuscript.
Additionally, within the file "Rosalia ESSD.html" under 9 GIS datasets the "Map of sites - Detail Q2" contains elements in the legend which are not part of the map.

R: changed and updated in the repository